# Soil microbial diversity and network complexity promote phosphorus transformation – A case of long-term mixed plantations of *Eucalyptus* and a nitrogen-fixing tree species

Jiyin Li [a, 1], Yeming You [a, b, 1], Wen Zhang [c], Yi Wang [d], Yuying Liang [a], Haimei Huang [a], Hailun Ma [a], Qinxia He [a], Angang Ming [b, e], Xueman Huang [a, b, *]

[a] Guangxi Key Laboratory of Forest Ecology and Conservation, Guangxi Colleges and Universities Key Laboratory for Cultivation and Utilization of Subtropical Forest Plantation, College of Forestry, Guangxi University, Nanning, Guangxi 530004, China

[b] Guangxi Youyiguan Forest Ecosystem National Observation and Research Station, Youyiguan Forest Ecosystem Observation and Research Station of Guangxi, Pingxiang 532600, Guangxi, China

[c] Jinggangshan Institute of Red Soil, Jiangxi Academy of Agricultural Sciences, Ji'an Jiangxi 343016, China

[d] Institute of Resources and Environment, Key Laboratory of National Forestry and Grassland Administration/Beijing for Bamboo & Rattan Science and Technology, International Centre for Bamboo and Rattan, Beijing, 100102, China

[e] Experimental Centre of Tropical Forestry, Chinese Academy of Forestry, Pingxiang 532600, China

[*] Corresponding author. Tel.: +86 13667881541; E-mail address: huangxm168168@163.com (X. Huang)

[1] These authors contributed equally to this work.

**Abstract**

Increased nitrogen (N) availability influences soil phosphorus (P) cycling through multiple pathways. Soil microorganisms are essential facilitating a wide range of ecosystem functions. However, the impact of mixed plantations of *Eucalyptus* and N-fixing tree species affect P transformation and microbiota interactions remains unknown. Therefore, we conducted a 17-year field experiment comparing pure *Eucalyptus* plantations (PPs) and mixed plantations (MPs) with *Eucalyptus* and a N-fixing tree species to assess their effects of soil P transformation, using data collected from two soil layers (0 –10 cm and 10–20 cm). The results showed that α-diversity indices (ACE and Chao1 and Shannon indices) were significantly higher in MPs than in PPs for both bacteria and fungi. Furthermore, MPs exhibited significantly higher relative abundances of bacterial phyla *Proteobacteria* (0–10 cm), *Verrucomicrobia*, and *Rokubacteria*, as well as fungal phyla *Mortierllomycota*, *Mucoromycota*, and *Rozellomycota*. Conversely, MPs showed lower abundances of the bacterial phyla *Chloroflexi*, *Actinobacteria*, and *Planctomycetes* and fungal phylum *Ascomycota*. Gene copy numbers of functional genes were also elevated in MPs, including 16S rRNA, internal transcribed spacer (ITS), N functional genes [*nifH* (0–10 cm), AOB-*amoA*, *narG*, *nirS*, and *nosZ* (0–10 cm)], and P functional genes [*phoC*, *phoD* (0–10 cm), *BPP*, and *pqqC*]. The findings indicated that MPs can enhance soil microbial diversity, network complexity, and the relative abundance of functional genes which involved N- and P- transformation by optimizing soil nutrient levels and pH, thereby facilitating P transformation. Therefore,

MPs of *Eucalyptus* and N-fixing tree species may represent a promising forest
management strategy to improve ecosystem P benefits.
**Keywords:** Co-occurrence network; functional gene; mixed plantation; N-fixing
species; phosphorous transformation

**1. Introduction**

Phosphorus (P) a vital macronutrient for plant and microbial growth (Turner et al., 2018), while the availability of P serves as a key indicator of soil fertility and quality (Peng et al., 2021). In most ecosystems particularly in tropical and subtropical forests, P bioavailable in soil is often limited due to intense weathering and the presence of aluminium (Al) ions and free iron (Fe) (Soltangheisi et al., 2019; Du et al., 2020). Therefore, these P reserves cannot be accessed directly by plants (Fan et al., 2019). However, plants and microorganisms have developed various strategies for access P from inorganic (Pi) and organic (Po) reservoirs and rendering it available for biological processes (including, e.g., assimilation by phosphate-solubilizing microorganisms and mineralization of enzymes) (Lu et al., 2022). Consequently, it is crucial to implement strategies for the sustainable management of soil P to enhance its utilization by plants, preserve soil quality, and mitigate the risk of P loss.

Soil microorganisms serve as both a reservoir and a source of phosphate ions, significantly influencing the availability of P. In addition, microorganisms play a role in maintaining soil functions such as nutrient cycling, biological activity, and plant growth, all of which are crucial for sustaining soil quality and fertility (Bünemann et al., 2008; Zhou et al., 2018; Sun et al., 2022). Microorganisms facilitate the P transformation by participating in the processes of P mineralization, solubilization, and cycling, converting P into bioavailable forms for plant uptake (Pastore et al., 2020). Specifically, the mineralization of Po is facilitated by the extracellular presence of phosphatases, which are mainly produced principally by soil microorganisms (Nannipieri et al., 2012). It is thus of both extracellular acid (ACP) and alkaline (ALP)

phosphatase activities are commonly used as the indicators to assess the
mineralization of Po to bioavailable Pi (Luo et al., 2019). Furthermore, P
transformation is influenced by the α-diversity, structure, and composition of soil
microbial communities, with pH being considered a key determinant in shaping
microbial diversity and community composition (Jin et al., 2019). Microbiome
co-occurrence networks are prevalently employed to scrutinize the interrelationships
within microbial communities, and network attributes (e.g., the mean degree, edge
quantity, and node amount) can be utilized to appraise the reciprocal ties among these
communities and their reactions to modifications in cultivation paradigms (Faust,
2021; Qiu et al., 2021). Microbial network analysis can uncover the complex
interactions between microorganisms, such as competition, cooperation, and
antagonism, while also shedding light on important ecological processes and
functional relationships that are not fully captured by microbial diversity analysis
alone. For instance, it can reveal processes like the transformation and cycling of key
soil nutrients (e.g., P and N), which are often overlooked in traditional diversity
assessments (Yao et al., 2024). Thus, gaining insight into the relationship between
microbial diversity, microbial network complexity, and the transformation and cycling
of P is crucial for improving soil functions and enhancing soil fertility.
The studies on genes involved in P cycling also emphasizes the contribution of
microbes in enhancing plant P uptake and efficiency (Dai et al., 2020). The P cycle
cluster includes genes that stimulate the mineralization of Po (e.g., *phoD*, *phoC*, and
*BPP*) (Cao et al., 2022; Khan et al., 2023) and solubilization of Pi (e.g., *pqqC*) (Meyer
et al., 2011). The genes *phoD*/*phoC* encode phosphatases, which are capable of
mineralizing Po compounds into Pi (Fraser et al., 2015). N is a fundamental element
for plant growth and development, typically coupled with P in biogeochemical cycles.
The N cycle group consists of genes responsible for microbially driven nitrification
(e.g., AOB-*amoA*), N fixation (e.g., *nifH*), and denitrification (e.g., *nirS*). Improved
interaction networks among soil microbial functional groups contribute to increasing
nutrient availability and enhancing the nutrient acquisition of host plants (Shi et al.,
2020; Qin et al., 2024). In addition, given that both N and P are essential elements for
microorganisms, an increase in N content can influence soil pH, which subsequently
alters the composition of soil microbial communities and impacts the abundance of
phosphatase-coding genes (*phoC* and *phoD*) (Widdig et al., 2020). Furthermore, the
presence of N-fixing plants also affects P uptake by enhancing litter decomposition
rates and the release of organic acids from microbial biomass, thereby accelerating
nutrient cycling and improving soil fertility (Li et al., 2021). Therefore, studying the
coupling of N and P cycling in soil is crucial for understanding of the diversity and
mechanisms of microbially driven biogeochemical cycles.
*Eucalyptus* is characterized by their straight trunks, well-developed horizontal root
systems, and high adaptability. They are prevalent in subtropical and tropical regions,
where they have significant economic and ecological value (Zhang and Wang, 2021).
However, monocultures and short-term rotation management of *Eucalyptus* plantation
have led to soil degradation, reductions in plant-available soil nutrient effectiveness
(e.g., the availability of nutrients such as N, and P in forms that can be absorbed and
utilized by plants), and soil microbial function and diversity, as well as other adverse

ecological effects. Mixed-species forests exert a strong positive impact on soil fertility and nutrient cycling by regulating the microbiome, including its diversity and structure (Pereira et al., 2019, Li et al., 2024). Recently, incorporating N-fixing trees species such as *Acacia* as a substitute for N fertilization has become widely acknowledged as one of the most effective silvicultural practices for enhancing tree N uptake and woody production in *Eucalyptus* plantations (Koutika and Mareschal, 2017; Zhang et al., 2023). In addition, mixing with N-fixing tree species improves N availability, P accumulation, microbial diversity, and forms a more complex and interconnected microbial network compared to pure plantations (Li et al., 2022; He et al., 2024; Yao et al., 2021). So far, the effect of N-fixing tree species on P cycling has mainly been addressed by investigating organic or inorganic P accumulation in soil from either pure or mixed stands of non - N-fixing tree species and N-fixing tree species (Yao et al., 2024).

*Acacia mangium,* one of the N-fixing trees species that is widely planted in many parts of the world, has clear benefits in forestry and agroforestry ecosystems (Epron et al., 2013; Koutika and Richardson, 2019). Key reasons for the widespread planting of *Acacia mangium* in pure or mixture plantations with other tree species with infertile soils, are its capacity to change soil faunal, microbial communities (Huang et al., 2014; Pereira et al., 2017), improve soil fertility (Tchichelle et al., 2017), and stimulate tree growth and forest productivity (Paula et al., 2015). Nevertheless, the effects of mixing N-fixing trees species on regulating the correlations between microbial diversity and network of P transformation is still poorly understood. Phosphomonoesterase (e.g., ACP) mineralization is an essential strategy for P transformation (Luo et al., 2019; Yu et al., 2022; Wang et al., 2023), so we employed soil ACP activity to analyse the dynamics of P transformation. Here, we aimed to (1) compare the variations in the

structure, diversity, and stability of soil microbial communities after mixing
*Eucalyptus* with N-fixing tree species, and (2) elucidate the mechanisms through
which fungal and bacterial communities, along with genes associated with N and P
transformation processes, regulate P transformation. We hypothesized that (1)
mixed-species plantations of *Eucalyptus* and N-fixing tree species would alter the
composition of soil microbial communities and improve microbial community
diversity and network complexity in the soil; (2) introduction of N-fixing tree species
may cause imbalance in soil properties (e.g., SOC, pH and so on), microbial diversity
and networks complexity, and related functional genes which co-regulated the P
transformation with differential roles. Our findings will provide more new insights
into sustainable management practices for plantations.
**2. Materials and methods**
*2.1. Site description*
The study was conducted in the Shaoping Experimental Field at the
Experimental Center for Tropical Forestry, which is affiliated with the Chinese
Academy of Forestry (106°56′E, 22°03′N). The area has a subtropical climate, with
approximately 1,400 mm of rainfall annually and maintaining an average yearly
temperature of 21.2°C. The landscape is characterized by low mountains and hills
along with acidic red soil. Forests in this area are primarily composed of
commercially managed plantations, as either pure or mixed stands.
*2.2. Plot design and sampling*
In this study, the pure (monoculture) *Eucalyptus urophylla* plantations (PPs) and
adjacent mixed plantations (MPs) of *Eucalyptus urophylla* and *Acacia mangium*
(N-fixing tree species) were established in 2004 on the logging tracks of *Pinus*
*massoniana* plantations that were established in 1977. The MPs were planted at a 1:1
mixing ratio with inter-row planting, consisting of one row of *Eucalyptus urophylla*
and one row of *Acacia mangium*. In the first two consecutive years post-planting, both
plantations were subjected to a similar stand management regime, which included
practices such as weed control and fertilization, subsequently allowing them to
proceed with their natural stand development. The experimental design is described in
the study conducted by Huang et al. (2017). In 2021, taking into account the
differences in plantation layout and topography, five 20 m × 20 m sample plots were
randomly established in each stand (PPs and MPs), ensuring that adjacent plots
maintained at a distance greater than 200 m to mitigate edge effects. The diameter at
breast height, height, and stand density of every tree within each plot were assessed.
Detailed information on the plantations is provided in Table A1.
Soil samples were carried out in early August 2021. Soil samples were gathered
from eight different points within each plot, located at 5 m intervals from the center,
along angles of 0°, 45°, 90°, 135°, 180°, 225°, 270°, and 315°. Previous studies only
examine a single soil layer (usually the upper 0–10 cm) (Waithaisong et al., 2022;
Chen et al., 2024). More study on the P transformation and mechanisms underlying
soil microbial and biochemical interactions in different soil layers is needed to
determine whether the variation of P cycle is dependent on depth. Therefore, soil
samples in our study were obtained from the depth intervals of 0−10 cm and 10−20
cm following the removal of extraneous materials such as little stones, and dead
leaves. Eight undisturbed samples from each soil layer were amalgamated into a
composite sample and transported to the laboratory on ice. Each composite sample
was partitioned into two aliquots: one designated for the analysis of physicochemical
properties, and the other reserved for genomic DNA extraction.
*2.3. Soil properties and soil enzyme activity*

Soil pH was measured using a 1:2.5 soil-to-water ratio method, and soil organic

carbon (SOC) was quantified using the $K_2Cr_2O_7$-$H_2SO_4$ oxidation method. The total

nitrogen (TN) content of soil was determined using an Auto Analyzer III in an extract

obtained by digestion of the sample with $H_2SO_4$ and a catalyst ($CuSO_4$:$H_2SO_4$ = 10:1).

The levels of nitrate N ($NO_3^-$-N) and ammonia N ($NH_4^+$-N) were determined by $CaCl_2$

extraction, followed by quantitative analysis using an AutoAnalyzer III (Tsiknia et al.,

2014). Total P (TP) was quantified using the molybdenum blue colorimetric method

following extraction of the samples with $HClO_4$-$H_2SO_4$ (Murphy and Riley, 1962).

N and P metabolismed by soil extracellular enzyme activity (EEA), e.g.,

β-1,4-N-acetylglucosaminidase (NAG) and leucine aminopeptidase (LAP) activity are

involved in N acquisition and acid phosphomonoesterase is associated with P

mineralization, were quantified in a fluorescence assay conducted in a 96-well

microplate (Yan et al., 2022). Soil EEA was calculated from the fluorescence readings

of the enzyme after its reaction with the appropriate substrate. The assay was

conducted using 200 µL of a soil suspension prepared by weighing 1.25 g of fresh soil

to which sodium acetate buffer (pH 4.5) was added, and stirred for 1 min to ensure

consistent extraction conditions and effective solubilization of the soil constituents.

Eight replicates per sample were tested. The samples were incubated in darkness at

25°C for 3 h, after which the reaction was terminated by adding NaOH. Fluorescence

was then immediately measured within the wavelength range of 365–450 nm by using

a fluorescence microplate reader. Information on the substrates of the three EEA can

be found in Table A2.

*2.4. Soil DNA extraction and sequencing*
Microbial genomic DNA was obtained from soil samples utilizing the PowerSoil
DNA isolation kit (MN NucleoSpin 96 Soi) for subsequent analysis and
measurements. The primers employed were 338F and 806R for the amplification of
the V3–V4 hypervariable region of the 16S rRNA gene (Mori et al., 2014; Parada et
al., 2016), while ITS1F and ITS2R were employed to amplify the ITS1 region of
fungal rRNA gene loci (Adams et al., 2013; Dong et al., 2021) (Table A3).
Sequencing data were processed by filtering the raw reads using Trimmomatic v0.33,
removing the primers using Cutadapt v1.9.1, assembling the clean reads by overlap
with Usearch v10, and removing chimeras with UCHIME v4.2 to ensure data validity.
After the removal of potential chimeras, 1,600,678 and 1,550,033 high-quality
bacterial and fungal reads were obtained, respectively.
The genetic potential of the soil microorganisms was assessed by real-time
fluorescence quantitative PCR (qPCR) to quantitatively determine the gene copy
numbers of bacteria (16S rRNA) and fungi (ITS). The genetic potential of N cycling
processes was evaluated based on the abundance of functional genes involved in
nitrogen fixation (*nifH*), nitrification (AOB-*amoA*), and denitrification (*narG*, *nirS*,
*nirK*, and *nosZ*). Similarly, the genetic potential of P cycling processes was assessed
using the abundance of functional genes involved in organic phosphorus hydrolysis
(*phoC*, *phoD*, *BPP*) and Pi hydrolysis (*pqqC*). These functional genes are
well-established biomarkers of the biochemical pathways essential for nutrient
cycling in various ecosystems. The qPCR amplification efficiencies ranged from 90%
to 110%. The primers and references for the functional genes are reported in Table
A3.
*2.5. Network construction*

238        Networks for bacteria and fungi were constructed by dividing the 20 samples

into four groups, consisting of two soil layers for PPs and MPs, respectively. First,
sample operational taxonomic units (OTUs) were filtered, discarding those that
appeared in fewer than three samples within each group (3 out of 5 replicates) (Hu et
al., 2023). OTUs with a relative abundance exceeding 1% in the bacterial and fungal
communities were selected for further correlation analysis (Fan et al., 2018). The
network was built according to thresholds of Pearson correlation coefficient > 0.6 and
$P < 0.05$, assessed using the *Hmisc* package in R v4.0.5. We adjusted the *P* values
according to the Hochberg false discovery rate test (Benjamini et al., 2006), with a
cut-off of adjusted $P < 0.05$. Network properties were computed utilizing the *igraph* R
package, and visualized using Gephi (https://gephi.org/). In all figures, bacterial and
fungal phyla exhibiting a relative abundance greater than 1% within the network are
represented by distinct colors.

251        Keystone species were identified by utilizing the connectivity within modules (Zi)

and between modules (Pi). Microorganisms were classified into four categories
depending on intra-module degree (Z-score) and participation coefficient (C-score)
thresholds, into network hubs, module hubs, connectors, and peripherals (Poudel et al.,
2016). Network hubs refer to nodes with a high degree of connectivity both globally
and within individual modules; module hubs are nodes with significant connectivity
restricted to a single module; connectors are nodes that facilitate strong connections
between different modules, and peripheral nodes are those with few connections to
other nodes (Poudel et al., 2016). Network hubs, module hubs, and connectors occupy
critical positions within the network and are classified as keystone topological
features. These characteristics are essential for sustaining the stability of microbial
communities (Delmas et al., 2019). Consequently, OTUs associated with these nodes
were designated as keystone species.
*2.6. Data analyses*
Microbial diversity (Shannon index) and richness (Chao1 and ACE), which were
both calculated using phyloseq with default setting by Mothur (v 1.30.2) software
(Schloss et al., 2009). Soil physicochemical properties, microbial community indices,
such as the ACE and Shannon and Chao1 indices, as well as functional genes and
enzyme activity, were analyzed in independent samples t-tests using SPSS v24.0. This
statistical approach was applied to evaluate differences attributable to stand type
(monoculture or mixed). Differences in soil microorganisms across stand types and
soil layers were analyzed using non-metric multidimensional scaling (NMDS) with
Bray–Curtis dissimilarity and analysis of similarity (ANOSIM), implemented using
the *vegan* package in R (Oksanen et al., 2013; Knowles et al., 2019). Random forest
analysis based on Pearson correlation analysis and the best multiple regression model
was used to evaluate the contributions of soil properties, microbial characteristics, and
functional genes involved in the N and P cycles to the variation in nitrogen and
phosphorus transformation enzyme activities, and to identify the major predictors
based on their importance. Computation and visualization were carried out in R
software (Jiao et al., 2020). Correlation analysis and visualization of soil properties,
microbial characteristics, and functional genes related to N and P cycling were
performed in Origin 2024. A redundancy analysis (RDA) was employed to explore the
multivariate    associations    between    soil    physicochemical    characteristics    and
microorganisms. The most important soil physicochemical properties affecting
bacterial and fungal phyla were identified in the RDA and visualized using CANOCO
v5. A partial least squares path model (PLS-PM) was constructed using R software to
assess the direct and indirect effects of mixed planting of *Eucalyptus* and *Acacia* on P
transformation. A PLS-PM can reveal causal connections between observed and latent
variables, and its superiority for small sample sizes has been demonstrated in
simulation studies, in which path modeling estimation was shown to be reliable
(Monecke and Leisch, 2012; Sanchez, 2013). The goodness-of-fit statistic was used to
assess the adequacy of the PLS-PM fit, with a value > 0.7 indicating good model fit
(Tenenhaus et al., 2004; Sanchez, 2013).
**3. Results**
*3.1. Soil properties*
Significant ($P < 0.05$) higher of SOC, TN, $NO_3^-$-N, C:P, N:P, and pH were
determined in both two investigated soil layers in MPs than those in PPs (Table 1);
however, TP (10–20 cm) was significantly lower in MPs than in PPs ($P < 0.05$ (Table

299    1).

**Table 1** Soil physicochemical properties in both 0−10 cm and 10−20 cm soil layers in PPs and
MPs.

| Soil physicochemical properties | Stand type | M±SE | t | *P* | M±SE | t | *P* |
|---|---|---|---|---|---|---|---|
| | | 0–10 cm | | | 10–20 cm | | |
| SOC | PP | 12.98±0.90b | -5.790 | *P* < 0.001 | 10.31±0.79b | -4.189 | *P* < 0.001 |
| | MP | 21.18±1.10a | | | 14.45±0.59a | | |
| TN | PP | 1.15±0.04b | -6.658 | *P < 0.001* | 0.83±0.02b | -5.551 | *P* < 0.001 |
| | MP | 2.17±0.15a | | | 1.33±0.09a | | |
| NH$_4^+$-N | PP | 18.92±1.49a | 1.402 | *P* < 0.001 | 13.84±0.83a | 2.262 | *P* = 0.001 |
| | MP | 15.14±2.25a | | | 11.71±0.44a | | |
| NO$_3^-$-N | PP | 4.86±0.06b | -13.372 | *P* = 0.198 | 3.05±0.05b | -33.443 | *P* = 0.054 |
| | MP | 13.90±0.67a | | | 5.39±0.05a | | |
| TP | PP | 0.31±0.02a | 0.520 | *P* < 0.001 | 0.32±0.03a | 3.458 | *P* < 0.001 |
| | MP | 0.30±0.02a | | | 0.22±0.01b | | |
| C:N | PP | 11.38±0.96a | 1.497 | *P* = 0.167 | 12.37±0.89a | 1.182 | *P* = 0.009 |
| | MP | 9.82±0.39a | | | 10.98±0.76a | | |
| C:P | PP | 42.04±3.18b | -4.887 | *P* = 0.173 | 32.73±2.47b | -8.865 | *P* = 0.271 |
| | MP | 72.75±5.35a | | | 64.63±2.62a | | |
| N:P | PP | 3.74±0.25b | -7.173 | *P* = 0.001 | 2.67±0.17b | -6.093 | *P* < 0.001 |
| | MP | 7.37±0.44a | | | 6.00±0.52a | | |
| pH | PP | 4.28±0.04b | -6.970 | *P* < 0.001 | 4.21±0.05b | -5.824 | *P* < 0.001 |
| | MP | 5.09±0.11a | | | 5.04±0.13a | | |

SOC: Soil Organic Carbon; TN: Total Nitrogen; NH$_4^+$-N: Ammonium Nitrogen; NO$_3^-$-N: Nitrate
Nitrogen; TP: Total Phosphorus; C:N: Carbon: Nitrogen ratio; C:P: Carbon: Phosphorus ratio; N:P:
Nitrogen: Phosphorus ratio; pH: Soil pH Value; Value = Mean ± Standard Error; Different
lowercase letters in the table represent significant differences between PPs and MPs (*P* < 0.05),
the same below.
*3.2. Bacterial and fungal community diversity and composition*

308        In both soil layers, the bacterial ACE (0–10 cm: t = -5.164, *P* = 0.001; 10-20 cm:

t = -7.305, *P* < 0.001), Chao1 (0–10 cm: t = -5.039, *P* = 0.001; 10-20 cm: t = -6.387, *P*
< 0.001), and Shannon (0–10 cm: t = -3.478, $P$ = 0.008; 10-20 cm: t = -3.772, $P$ <
0.005) indices of α-diversity were significantly higher in MPs than in PPs (Fig. 1a–c).
Fungal Shannon (t = -3659, $P$ = 0.006) index in the 0–10 cm was also significantly
higher in MPs than in PPs (Fig. 1f). The composition of bacterial and fungal
community exhibited significant differences between the two plantation types and soil
layers, except for the fungal communities in PPs, which did not differ between the
surface and deeper soil layers ($P$ < 0.05, ANOSIM: $R^2$ = 0.85, $P$ = 0.01, stress = 0.03
and $R^2$ = 0.73, $P$ = 0.01, stress = 0.05, respectively, Fig. A1).

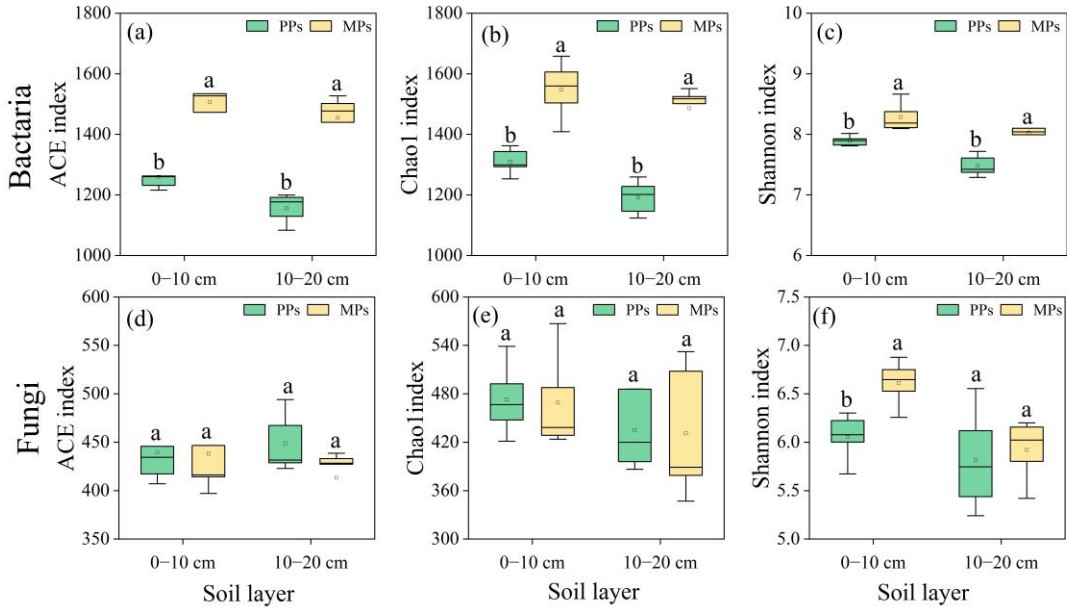


**Fig. 1** Comparisions of (a-c) bacterial and (d-f) fungal community, by α diversity index in two soil
layers in PPs and MPs. Different lowercase letters in the table represent significant differences
between PPs and MPs ($P$ < 0.05), the same below.

322        After clustering at a 97.0% similarity level, a total of 1,869 OTUs were

obtained for bacteria, which revealed 21 phyla, 64 classes, 140 orders, 201 families,
and 311 genera. For fungi, a total of 1,128 OTUs were obtained, showing 8 phyla, 24

classes, 62 orders, 104 families, and 157 genera (Table A4). The most abundant bacterial phyla (relative abundance > 1%) in both PPs and MPs were *Acidobacteria* (26.83%), *Proteobacteria* (22.46%), *Chloroflexi* (13.95%), *Actinobacteria* (13.62%), *Verrucomicrobia* (11.16%), *Planctomycetes* (5.6%), and *Rokubacteria* (3.5%), which represented 94.08% of the total bacterial community in the 0–10 cm layer (Figs. 2a, b and A2a). The most abundant fungal phyla (relative abundance >1%) in both PPs and MPs were *Ascomycota* (63.25%), *Basidiomycota* (28.14%), *Mortierellomycota* (1.77%), *Mucoromycota* (1.18%), and *Rozellomycota* (1.06%), which represented 95.40% of the total fungal community (Figs. 2c, d and A2b). The introduction of N-fixing tree species resulted in changes in the relative abundance and composition of these microbial communities, although these changes were not always statistically significant (Fig. 2).

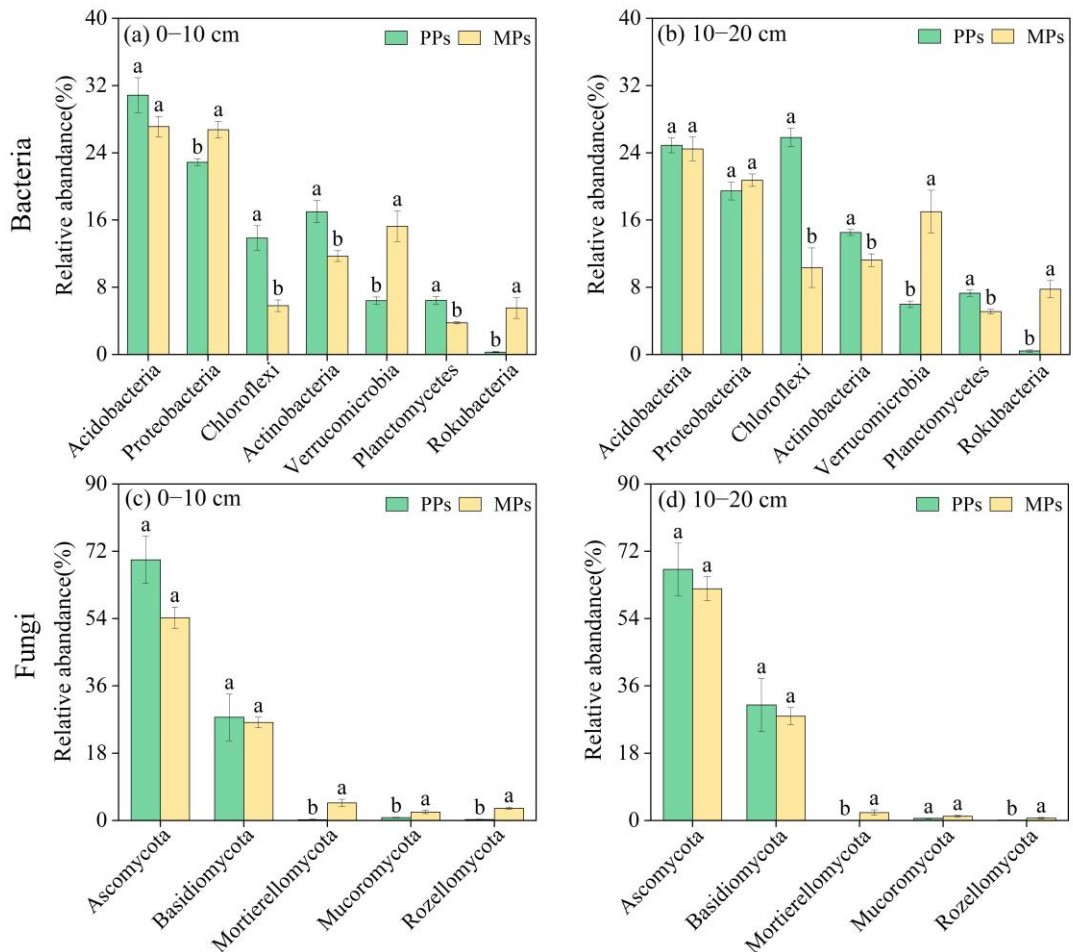

**Fig. 2** Abundance difference of (a-b) bacterial and (c-d) fungal and based on relative abundance > 1% at phylum level.

We used RDA to determine the linkage between soil microbial phyla and the specific soil physicochemical factors. The first two components of RDA axes explained 80.87% and and 47.75% of the total variance in the relationship between soil bacterial and fungal communities and nine selective soil physicochemical factors, respectively (Fig. 3a, b). Forward selection of the nine soil physicochemical factors in the RDA ordinations showed that the bacterial communities were primarily influenced by pH, TN, and SOC (Fig. 3a), and the fungal communities were primarily influenced by pH ($P < 0.05$) (Fig. 3b).

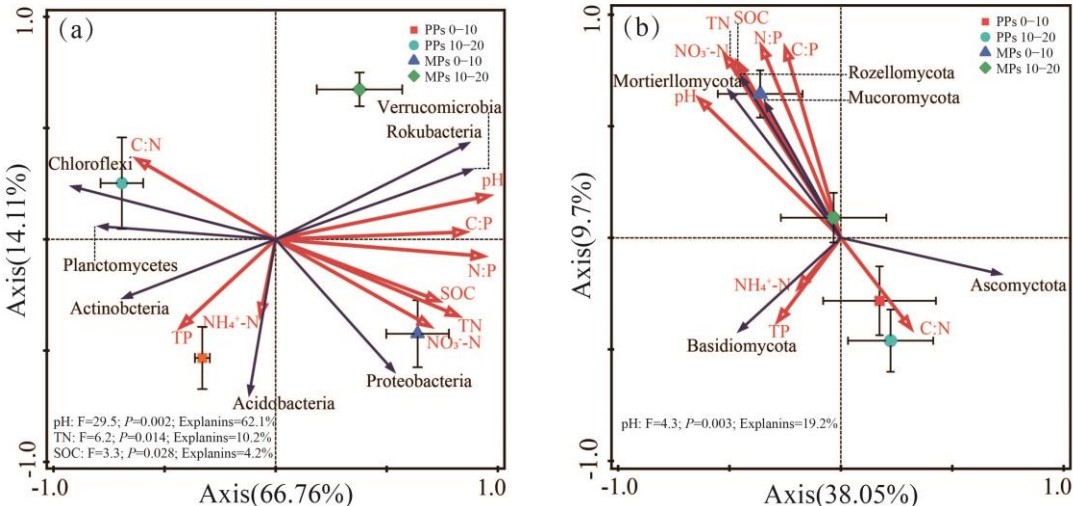

**Fig. 3** RDA plot showing significant factors affecting bacterial (a) and fungal (b) communities.

*3.3. Microbial network complexity and stability*

Microbial species with an average abundance of at least 1% in the 0–10 and 10–20 cm of PPs and MPs were selected for network analysis. Significant differences in microbial network structure were found between PPs and MPs in both soil layers (Fig. 4a, b). In the bacterial and fungal networks, there were significantly more nodes in MPs than in PPs (Table 2). Therefore, compared to PPs, MPs significantly stimulated the complexity of the co-occurrence network, particularly in the 0–10 cm. Positive correlations (bacterial networks: ranging = 0.665–0.712, fungal networks: ranging = 0.754–0.849) were determined for both PPs and MPs (Table 2). Compared with PPs, the average path lengths in MPs were shorter (except for the fungal network in the 10–20 cm) and the network diameter was smaller (except for the bacterial network in the 10–20 cm) and had a higher average degree for both the bacterial and the fungal networks in both soil layers (Table 2).

The Zi–Pi plot showed that network hubs were absent from the bacterial and fungal networks, with keystone species instead concentrated in connectors and

module hubs (Fig. 4c, d). Bacterial keystone OTUs were primarily found in the top
three phyla, *Proteobacteria*, *Acidobacteriota*, and *Actinobacteriota* (Fig. 4c). Fungal
keystone OTUs were likewise concentrated in the top three phyla, *Ascomycota*,
*Basidiomycota*, and *Mucoromycota* (Fig. 4d).
**Table 2** Co-occurrence network parameters of bacterial and fungal community at OTU level

| Species type | Soil layer (cm) | Stand type | Number of nodes | Number of edges | positive edges | negative edges | Average path length | Network diameter | Average degree |
|---|---|---|---|---|---|---|---|---|---|
| Bacteria | 0−10 | PPs | 529 | 2498 | 1661 | 837 | 13.58 | 38 | 9.44 |
| | | MPs | 667 | 7930 | 5403 | 2527 | 7.79 | 26 | 23.67 |
| | 10−20 | PPs | 447 | 2509 | 1786 | 723 | 9.41 | 27 | 11.23 |
| | | MPs | 581 | 6342 | 4257 | 2085 | 8.51 | 30 | 21.83 |
| Fungi | 0−10 | PPs | 298 | 642 | 484 | 158 | 6.47 | 22 | 4.31 |
| | | MPs | 344 | 859 | 722 | 137 | 5.80 | 20 | 4.99 |
| | 10−20 | PPs | 260 | 511 | 421 | 90 | 3.00 | 12 | 3.93 |
| | | MPs | 304 | 779 | 661 | 118 | 5.04 | 15 | 5.13 |


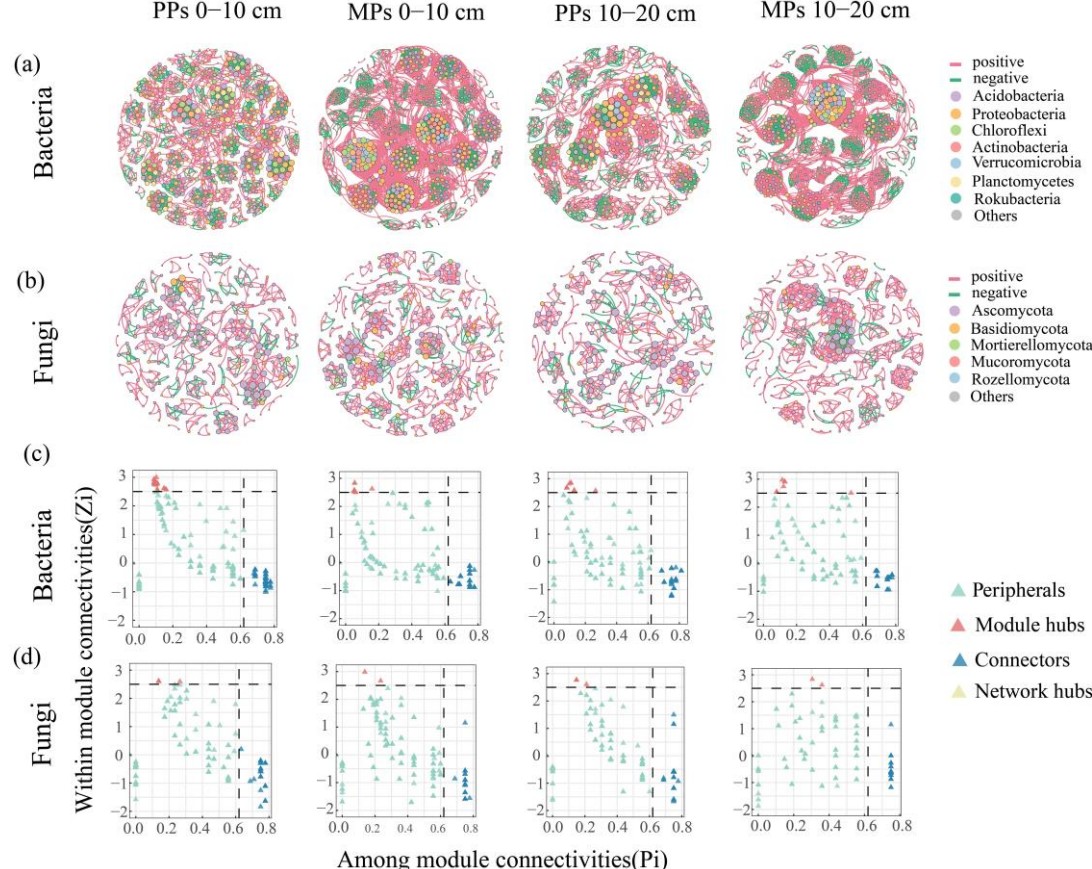

**Fig. 4** Co-occurrence network characteristics of (a) bacterial and (b) fungal communities. The node color represents the phyla with relative abundance greater than 1%, and the node size represents the degree. The Zi-Pi plot (c-d) predicts keystone OTUs in (c) bacterial and (d) fungal networks.

*3.4. Microbial functional genes involved in N and P transformation and enzyme activity*

Introducing *Acacia mangium* into the *Eucalyptus urophylla* plantation increased the abundances of functional genes involved in N and P transformation (Figs. 5 and 6). Specifically, the abundances of the N-related functional genes *nifH* (t = -4.218, *P* = 0.003), AOB-*amoA* (t = -3.648, *P* = 0.003), *narG* (t = -2.518, *P* = 0.036), *nirS* (t = -3.876, *P* = 0.005), and *nosZ* (t = -2.613, *P* = 0.031) in the 0–10 cm and of

AOB-*amoA* (t = -2.466, *P* = 0.039), *narG* (t = -2.482, *P* = 0.038), and *nirS* (t = -4.477,
*P* = 0.002) in the 10–20 cm were significantly higher in MPs than in PPs (Fig. 5a–f).
The abundances of the P functional genes *phoC* (0–10 cm: t = -4.316, *P* = 0.003;
10–20 cm: t = -4.177, *P* = 0.003), *phoD* (0–10 cm: t = -2.906, *P* = 0.020), *BPP* (0–10
cm: t = -6.373, *P* < 0.001; 10–20 cm: t = -2.956, *P* = 0.018), and *pqqC* (0–10 cm: t =
-3.746, *P* = 0.006; 10–20 cm: t = -4.403, *P* = 0.002) in both soil layers were
significantly higher in MPs than in PPs, with the exception of *phoD* in the 10–20 (Fig.

390    6).

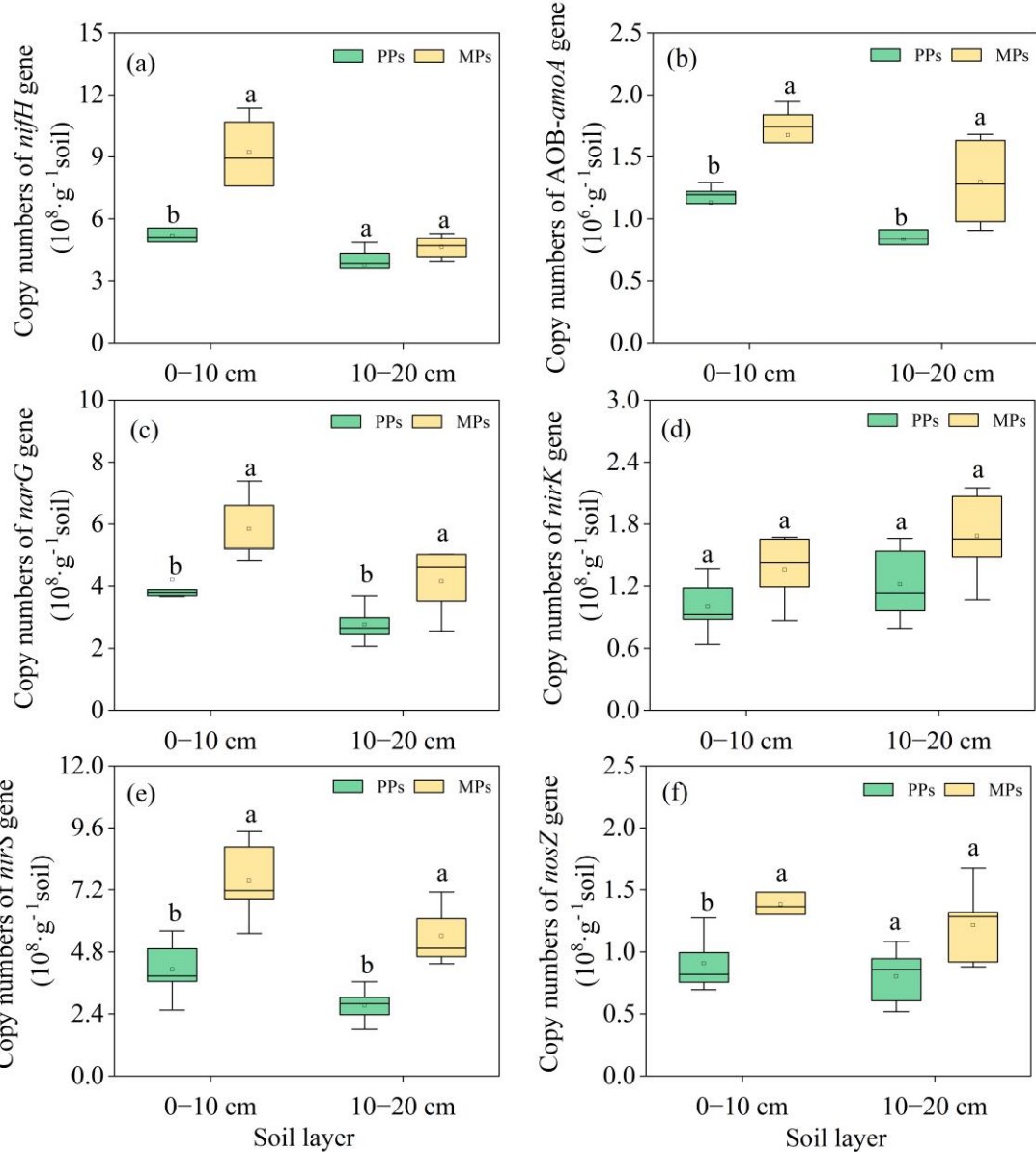


**Fig. 5** Comparison of the abundance of functional genes involved in nitrogen fixation (*nifH*) (a),

nitrification (AOB-*amoA*) (b), and denitrification [*narG* (c), *nirK* (d), *nirS* (e), and *nosZ* (f)] in two

soil layers in PPs and MPs.

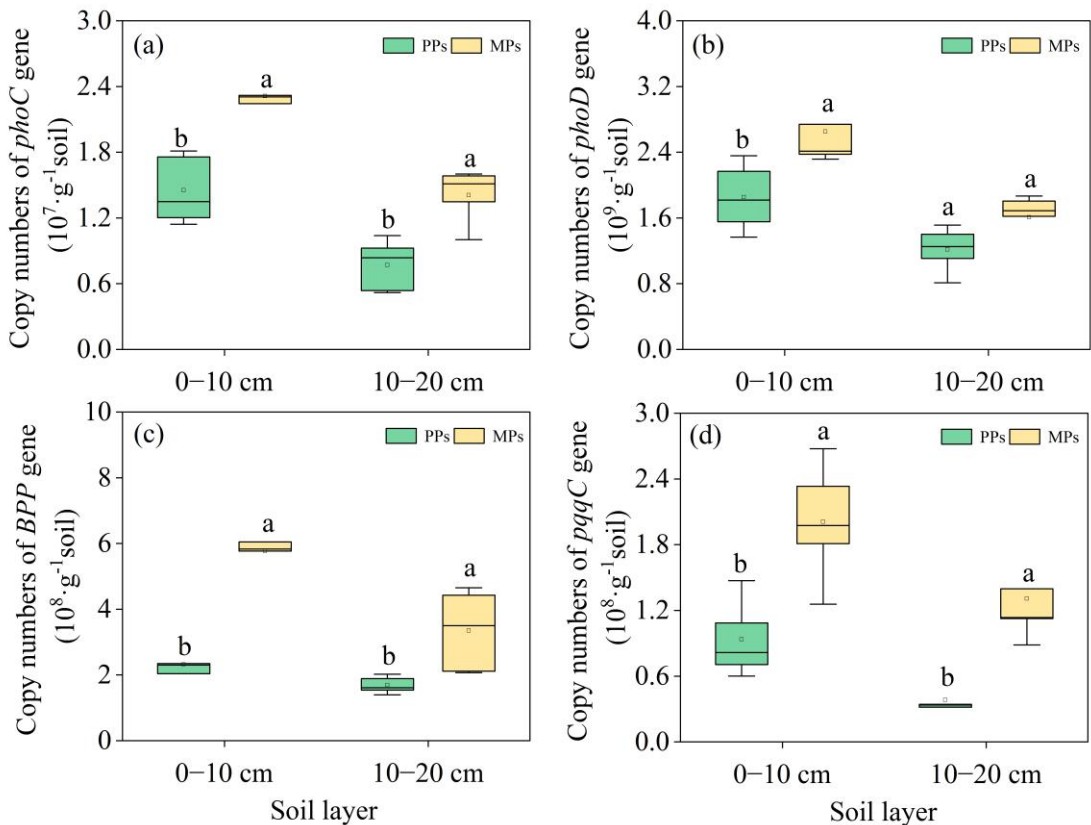

**Fig. 6** Comparison of the abundance of functional genes involved in Po hydrolysis [*phoC* (a), *phoD* (b), *BPP* (c)] and Pi hydrolysis (*pqqC)* (e) in two layers in PPs and MPs.

The EEA analysis results showed that NAG (t = -13.435, *P* < 0.001), LAP (t = -2.528, *P* = 0.035), and ACP (t = -5.291, *P* = 0.001) in the 0–10 cm were significantly higher in MPs than in PPs, by 97.31%, 31.72%, and 64.35% respectively (Fig. 7). In the 10–20 cm, NAG (t = -13.435, *P* < 0.001), LAP (t = -3.239, *P* = 0.012), and ACP (t = -4.102, *P* = 0.003) were also significantly higher in MPs than in PPs, by 24.02%, 88.54%, 39.83%, and 47.72%, respectively (Fig. 7). The qPCR results showed significantly higher levels of 16S rRNA (0–10 cm: t = -7.258, *P* < 0.001; 10–20 cm: t = -4.489, *P* = 0.002) and ITS (0–10 cm: t = -10.262, *P* < 0.001; 10–20 cm: t = -5.391, *P* = 0.001) in MPs than in PPs (Fig. A3).

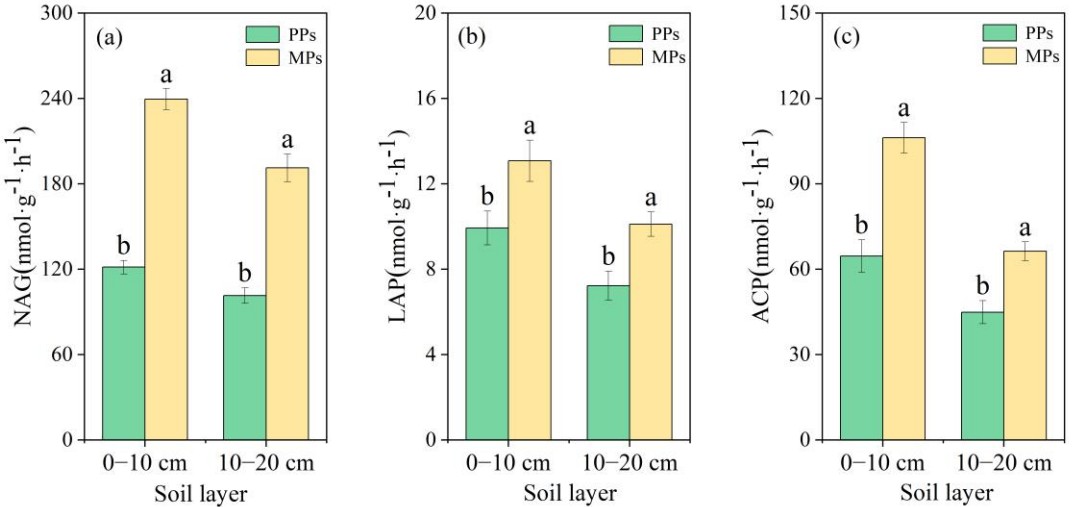

**Fig.7** Comparisions extracellular soil enzyme activity of (a) β-1,4-N-acetylglucosaminidase for chitin degradation (NAG); (b) Leucine aminopeptidase for protein degradation, (LAP); and (c) Acid phosphatase for catalyzing the hydrolysis of phosphate monoesters, ACP in two soil layers in PPs and MPs.

*3.5. Integrating variation in microbial diversity and network complexity with P transformation*

The random forest analysis results showed that NAG, LAP, and ACP activities were explained by soil properties, microbial characteristics, and functional genes involved in the N and P cycles to 84.09%, 58.95%, and 75.51%, respectively (Fig. 8). The results showed significant positive correlations for NAG, LAP, and ACP with SOC, TN, $NO_3^-$-N, C:P, N:P, and pH; for the three enzymes with 16S rRNA, $ACE_{bacteria}$, $Chao1_{bacteria}$, $Shannon_{bacteria}$, $nodes_{bacteria}$, $edges_{bacteria}$, and average $degree_{bacteria}$ ($P < 0.05$); for NAG, LAP, and ACP with ITS, $Shannon_{fungi}$, $edges_{fungi}$, and average $degree_{fungi}$; for LAP and ACP with $nodes_{fungi}$; for NAG, LAP, and ACP with *nifH*, *AOB-amoA*, *narG*, and *nirS*; for NAG and LAP with *nosZ*; and for NAG, LAP, and ACP with *phoC*, *phoD*, *BPP*, and *pqqC* (all $P < 0.05$). In addition, NAG was

significantly negatively correlated with average path length$_{\text{bacteria}}$ ($P < 0.05$). Soil
physicochemical properties (SOC, TN, NO$_3^-$-N), bacterial community diversity and
network complexity, as well as functional genes involved in the N (*nifH*) and P (*phoC*)
cycles are strong positive predictors of the variation in EEA.

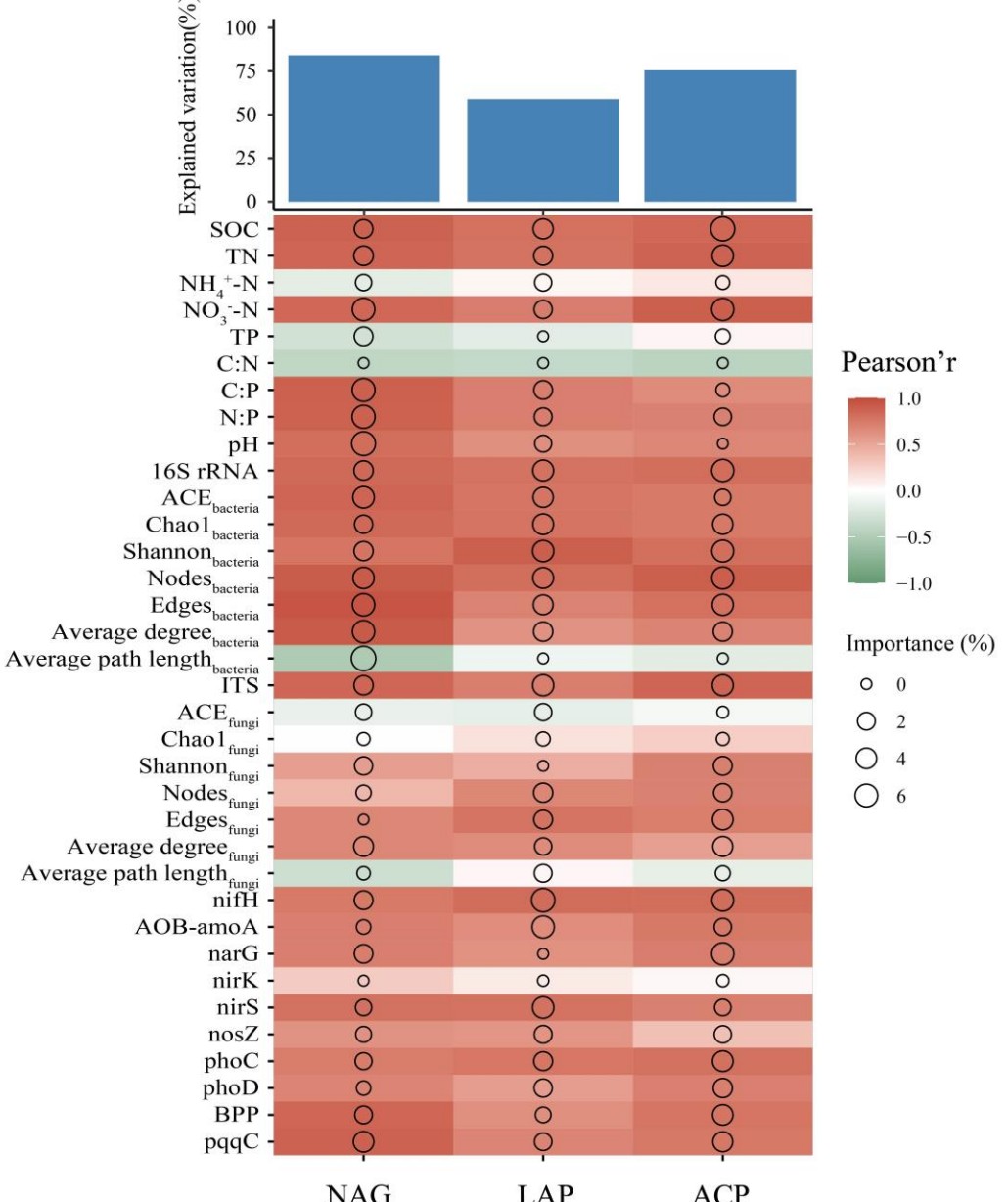


**Fig. 8** The potential biological contributions of soil properties, microbial influences, and

functional genes related to N and P cycling to the activity of N and P transformation enzymes. The
size of the circles represents the importance of the variables, and the color indicates the Pearson
correlation.

433       In the model of P transformation, the variance of 75.7%, 71.5%, 96.1%, 83.9%,

76.2 and 69.5% could be explained by soil properties, fungal properties, bacterial
properties, N functional genes, P functional genes, and N transformation, respectively,
within a goodness-of-fit index of 0.782 (Fig. 9a). N transformation and P functional
genes (*phoC*, *phoD*, and *BPP*) had a strong direct influence on P transformation, with
path coefficients of 0.283 and 0.605, respectively ($P < 0.01$). The diversity and
complexity of the network also had favorable effects on N and P functional genes,
exerting a substantial influence on P transformation. The overall influence of each
factor on P transformation in soil followed the order: soil properties > P functional
genes > bacterial properties > N functional genes > fungal properties > N
transformation (Fig. 9b). Overall, the mixture of *Eucalyptus* with N-fixing tree species
directly induces alterations in soil properties, which subsequently influence soil
microbial characteristics, functional genes involved in N and P cycling, as well as P
transformation, ultimately regulating P transformation.

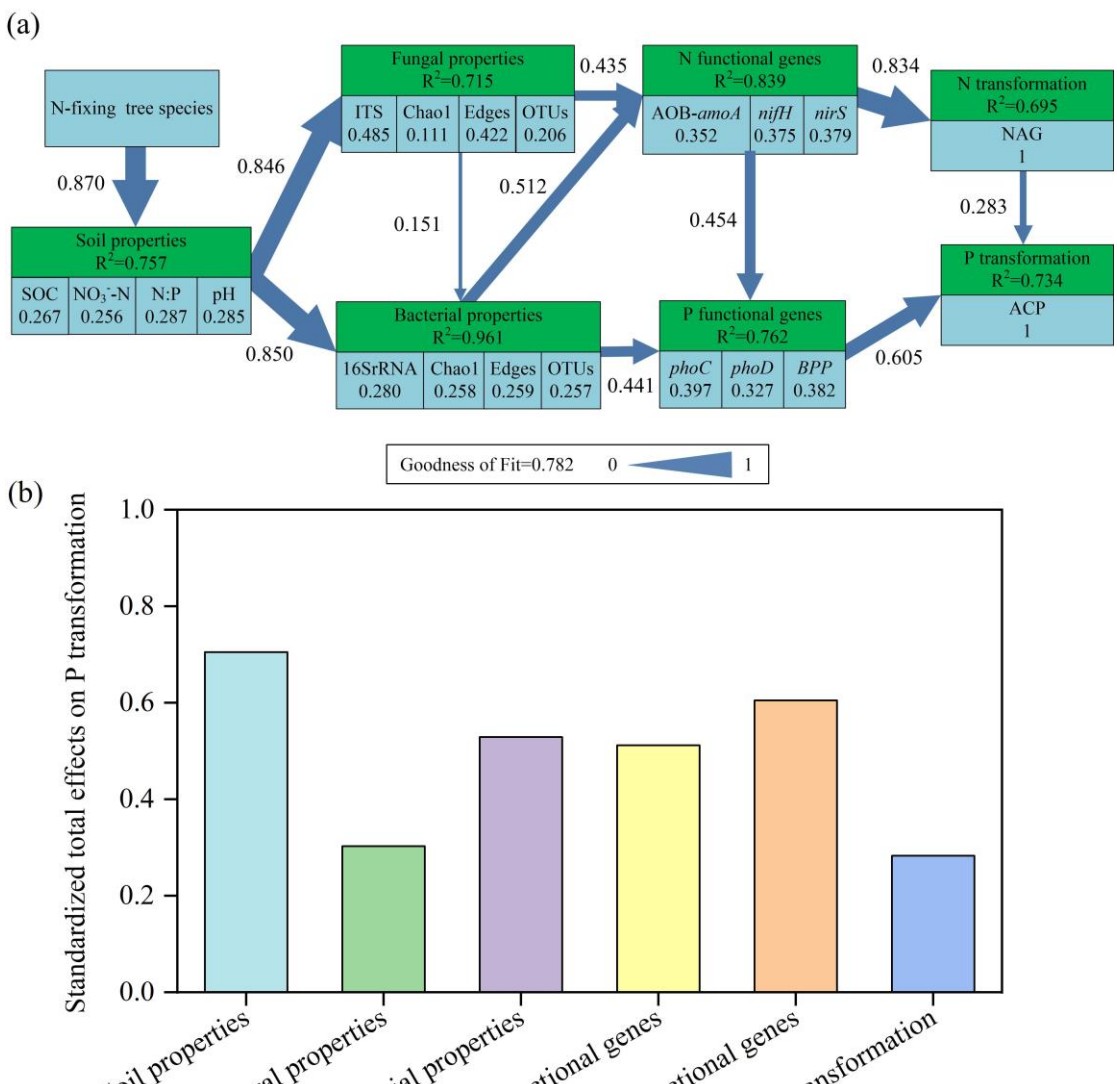


**Fig. 9** (a) Path model describing the control pathways of P transformation (ACP activity) and (b)

Standardized total effects (including both direct and indirect effects) on P transformation derived

from PLS-PM. The light blue in (a) represents the observation variable, the light green represents

the latent variable, the number under the observation variable represents the contribution weight of

the observation variable to the latent variable, the number and the width of the arrow on the arrow

represent the standardized path coefficient between the latent variables, and $R^2$ represents the

explanation rate of the model to the latent variable.

## 4. Discussion

*4.1 Soil microbial diversity and network response in a mixed plantation of Eucalyptus and N-fixing tree species*

The mixed planting of *Eucalyptus* with N-fixing species significantly impacted the soil microbial community structure, increasing microbial diversity and network complexity. With methodological advances that enable more comprehensive understanding of soil microbial diversity and network, we know that soil microorganisms are not only involved in nutrient (e.g., N and P) transformations but also shape the soil habitat by multiple biophysical and biogeochemical processes (Philippot et al., 2024). In our study, the combination of *Eucalyptus* and N-fixing *Acacia mangium* enhanced soil nutrient content and altered the stoichiometric ratios of C, N, and P (Table 1). Mixed plantations with N-fixing tree species have higher litter quantity and quality, which enhances nutrient retention and acquisition capacity (Huang et al., 2014), stimulates microbial growth, and promotes microbial aggregation and metabolism, thereby increasing microbial diversity (Guo et al., 2019) (Figs. 1 and A1). These findings align with those of a previous study, which demonstrated that the incorporation of *Eucalyptus* with N-fixing tree species increased the abundance and diversity of microorganisms, while also revealing variability in community structure across different stands (Li et al., 2023). The composition and diversity of soil microbial communities are primarily driven by C:N:P ratios (Delgado-Baquerizo et al., 2017). The availability of essential nutrients such as N, P, and Fe are controlled by the soil C supply, while the lower C:N ratio in

mixed plantations promotes the formation of various C components, thereby
increasing SOC input, which subsequently influences the structure of the microbial
communities and their co-occurrence patterns (Yuste et al., 2011; Qiu et al., 2021).
Interestingly, in this study, the TP content in MPs was significantly lower than that in
PPs (Table 1), which may be a result of increased plant uptake due to higher biomass.
Additionally, the high soil N content in MPs with N-fixing tree species may positively
influence plant growth, potentially stimulating P uptake (Li et al., 2016). In
subtropical regions, characterized by high temperatures and heavy rainfall, P leaching
is substantial; however, the introduction of N-fixing tree species increases N content,
which may shift the limitation from N to P in MPs. In this context, plants are likely to
recycle P more efficiently (See et al., 2015; Lang et al., 2016). Therefore, P returned
to the soil through decomposition would be reduced.
In natural habitats, soil microbial communities form intricate arrays and robustly
structured networks that allow adaptation to shifting environments (de Vries et al.,
2018). The complexity and diversity of microbial communities in soil are fundamental
to ecosystem persistence and resilience, as they both reinforce ecological functions
and offer a robust defense against external disruptions (Guo et al., 2021). The
complexity of the topological structure and connectivity between nodes influence the
overall stability of microbial networks and their resilience to environmental
disturbances (Yuan et al., 2021). The overwhelming predominance of positive over
negative correlations indicated microbial adaptation to similar ecological niches
through co-operation (Gao et al., 2022). Networks characterized by higher

connectivity and larger numbers of interrelationships are better equipped to withstand environmental changes, thereby preserving the functional stability of the ecosystem (Cornell et al., 2023). Our study showed that N-fixing tree species mixed plantations increased the complexity of bacterial and fungal networks (Fig. 4), as demonstrated by a higher number of nodes and edges, with positive associations predominating over negative ones, indicating stronger interactions between microorganisms (Ma et al., 2020; Niraula, 2021). Random forest analysis also revealed a robust positive association between the number of nodes and the diversity of fungal and bacterial species expressing enzymes responsible for N and P transformation (Fig. 8). These results align with our hypothesis, suggesting that *Eucalyptus* mixed with N-fixing tree species increases the complexity of microbial networks (Guo and Gong, 2024). The relative abundances of *Proteobacteria*, *Rokubacteria*, and *Verrucomicrobia* in the bacterial community were also higher in MPs than in PPs (particularly in the 0–10 cm), as were the relative abundances of *Mortierllomycota*, *Mucoromycota*, and *Rozellomycota* in the fungal community. Several edaphic factors collectively influenced the structure of both communities, among which pH was the most important (Fig. 3a, b). These findings are in line with earlier research, which demonstrated that soil pH was a key determinant in shaping the structure and composition of microbial communities (Siciliano et al., 2014; Cheng et al., 2020). According to our Zi–Pi plots, the keystone species of the bacterial community were members of phyla *Proteobacteria*, *Acidobacteriota*, and *Actinobacteria*, and those of the fungal community belonged to *Ascomycota*, *Basidiomycota*, and *Mucoromycota*.

The ability of leguminous plant species to establish symbiotic associations with root nodule bacteria, commonly referred to as rhizobia, is well established (e.g., Stougaard, 2000; Yang et al., 2022). The phylum *Proteobacteria* is one of the largest and phenotypically most diverse divisions, which includes gram-negative bacteria such as rhizobia. Furthermore, the N-fixing ability of rhizobia in the phylum *Proteobacteria* is a key contributor to maintaining the complexity and stability of microbial networks (Sprent and Platzmann, 2001; Fu et al., 2022). Among fungi, *Ascomycota* is the dominant phylum in soil worldwide (Egidi et al., 2019). In the present study, the relative abundance of Ascomycetes showed dominance in both PPs and MPs, but the relative abundance diminished in MPs. Although keystone taxa may not always abundant, they play a vital role in shaping microbial communities and maintaining their ecological functions, through specific regulatory pathways that affect community structure and function (Banerjee et al., 2018; Liu et al., 2022). For example, a prior study demonstrated that keystone taxa played a critical role in increasing the complexity of microbial networks, enhancing plant health and biomass, and promoting the hydrolysis of organophosphorus compounds through enzymatic activity (Qiao et al., 2024; Zeng et al., 2024).

*4.2 Association of microbial diversity and networks with P transformation and key environmental drivers*

Our study showed that the abundance of functional genes related to N and P cycles significantly increases after intercropping with N-fixing tree species, which supports our second hypothesis (Fig. 5 and 6). In contrast to this finding, Qin et al.

(2024) reported that although planting N-fixing tree species with *Eucalyptus* enhanced
the complexity and stability of N and P functional gene networks, it reduced the
abundances of these genes. This discrepancy can be explained by shifts in soil
microbial communities related to N and P cycles, which consequently affect the
microbial functions that respond to environmental changes (Graham et al., 2016;
Zhang et al., 2021). A previous study also found that the microbial community
associated with a mixed plantation of *Eurograndis* and *Amangium* differed from that
associated with monocultures of either species, attributable to positive effects of the
mixture on soil P and nitrate levels, which enhanced the abundances of N and P
functional genes (Rachid et al., 2013).
Biological N fixation is a fundamental ecosystem process that involves the
conversion of atmospheric N into a form usable by plants, which, facilitated by a
highly diverse group of microorganisms, significantly enhances soil fertility and
promoting plant growth (Burns and Hardy, 2012; Soumare et al., 2020). All N-fixing
microorganisms carry functional *nifH* genes that encode a component of nitrogenase
and act as markers of the abundance and diversity of N-fixing microorganisms across
various environmental contexts (Wang et al., 2018). Our results indicate that the
relative abundance of P functional genes was significantly higher after the
introduction of N-fixing tree species compared to pure *Eucalyptus* plantations (Fig. 6).
Both *phoC* and *phoD* are functional genes that encode phosphatase activity needed for
P solubilization and mineralization and are thus critically involved in promoting soil P
availability (Tian et al., 2021; Cao et al., 2022). The P cycling gene *pqqC*, which

encodes the P-mobilizing enzyme pyrroloquinoline quinone synthase, is a marker of phosphate-mobilizing bacteria (Meyer et al., 2011). The predominant bacteria containing *phoD* and *pqqC* are primarily members of the *Actinobacteria* and *Proteobacteria* (Tan et al., 2013; Hu et al., 2018), whose community structure was shown to remain unchanged with an increase in soil P pools (Ragot et al., 2015). In line with our results, a higher abundance and diversity of *phoD*-, *phoC*-, and *pqqC*-bearing soil microorganisms; higher abundances of these genes in soil were correlated with higher soil SOC and TN contents (Luo et al., 2019; Cao et al., 2022). Our study also identified significantly positive correlations between most N and P functional genes and 16S rRNA as well as the ACE, Chao1, and Shannon indexes in bacterial communities, whereas a significant positive correlation was determined only between the ITS region and the Shannon index in fungal communities (Fig. A4). This variation can be attributed to the significant positive impact that high levels of available nutrients have on the development of bacterial communities in the soil (Ming et al., 2016).

The significant positive correlations detected for the N enzymes NAG and LAP with AOB-*amoA*, *nifH*, and the denitrification genes *nirS*, *nosZ*, and *narG* determined in our study suggest that, after the introduction of N-fixing tree species, the microbial community facilitated soil N transformation by increasing the abundance of N cycling genes. Both random forest analysis and PLS-PM analyses indicated that P transformation reflected the interaction of biological and non-biological factors in ecological processes influenced by the introduction of

N-fixing tree species (Figs. 8 and 9). Complex interactions between bacteria, fungi,
and P cycle genes have been shown to promote microbial community stability while
facilitating P transformation processes (Liu et al., 2024). *Eucalyptus* mixed with
N-fixing tree species also increased soil TN and the $NH_4^+$-N content, which increased
ACP activity and thus soil Po mineralization. The higher soil pH in MPs than in PPs
was likely driven by exchange interactions involving Fe/Al hydroxide minerals and
functional groups (Table 1), which enhanced the conversion of potentially labile Pi
into plant available P via competitive adsorption (Hinsinger, 2001; Kang et al., 2021).
Together, these results indicate that forest management practices that
*Eucalyptus* mixed with N-fixing tree species will improve soil physicochemical
properties, microbial community diversity, and correlations between microbial N and
P cycling genes, thereby promoting soil P transformation.
**5. Conclusions**
This study suggests the benefits of incorporating mixed N-fixing tree species
with *Eucalyptus*, specifically highlighting their positive effects on P transformation.
The presence of *Acacia* was shown to alter soil physicochemical properties, improved
soil bacterial and fungal community diversity, network complexity, and the abundance
of N and P cycling functional genes, ultimately driving P transformation. Increases in
soil nutrient content, particularly SOC, TN, and $NO_3^-$-N, as well as the increase in pH
that occurred in MPs influenced soil microbial diversity. PLS-PM analysis revealed
that mixed plantations have significantly enhanced correlations between P
transformation and microbial functional genes that mediate N and P cycling. Our
findings offer fresh insights into the predictive capacity of potential shifts in the
belowground microbial communities for soil functionality within mixed plantation
ecosystems involving N-fixing tree species and *Eucalyptus*.

## Appendix A

**Table A1** Main characteristics in PPs and MPs.

| Stand type | Altitude (m) | Gradient (°) | Age (a) | SD (trees·hm⁻²) | DBH (cm) | TH (m) |
|---|---|---|---|---|---|---|
| PPs | 224 | 24 | 17 | 595±28 | 20.11±0.27 | 23.88±0.38 |
| MPs | 227 | 21 | 17 | 610±12 | 19.61±0.50 | 23.16±0.47 |
| *Eucalyptus urophylla* | — | — | — | 310±17 | 22.26±0.28 | 25.83±0.40 |
| *Acacia mangium* | — | — | — | 300±18 | 16.13±1.20 | 19.62±0.65 |

PPs: pure plantations; MPs: mixed plantations; SD: stand density; D.B.H.: diameter at breast height; TH: tree height.

**Table A2**. Details of the various soil extracellular enzymes and associated substrates.

| Enzyme Type | Enzyme | International Classification Number | Abbreviation | Substrate |
|---|---|---|---|---|
| N-acquiring enzyme | β-1,4-N-acetylglucosa minidase | EC 3.2.1.30 | NAG | 4-MUB-N-acetyl-β-D-glucosa minide (200 μM) |
| | Leucine aminopeptidase | EC 3.4.11.1 | LAP | L-Leucine-7-amino-4-methylc oumarin (200 μM) |
| P-acquiring enzyme | Acid phosphatase | EC 3.1.3.2 | ACP | 4-MUB-phosphate (200 μM) |

EC: Enzyme Commission number describing enzymatic function in increasing level of detail (the first number distinguishes 1-oxireductases, 2-transferases, 3- hydrolases, 4-lyases, 5-isomerases, and 6-ligases)

**Table A3** Quantitative real-time PCR primers for nitrogen and phosphorus cycling function genes.

| Gene type | Target gene | Primer | Sequence (5'- 3') |
|---|---|---|---|
| Nitrogen cycle | *nifH* | Pol-F | TGCGAYCCSAARGCBGACTC |
| | | Pol-R | ATSGCCATCATYTCRCCGGA |
| | *AOB-amoA* | amoA-1F | GGGGTTTCTACTGGTGGT |
| | | amoA-2R | CCCCTCKGSAAAGCCTTCTTC |
| | *narG* | narG-f | TAYGTSGGGCAGGARAAACTG |
| | | narG-r | CGTAGAAGAAGCTGGTGCTGT |
| | *nirK* | nirk876 | ATYGGCGGVCAYGGCGA |
| | | nirk1040 | GCCTCGATCAGRTTRTGGTT |
| | *nirS* | Nirs-Cd3aF | GTSAACGTSAAGGARACSGG |
| | | Nirs-R3cdR | GASTTCGGRTGSGTCTTGA |
| | *nosZ* | nosZ2F | CGCRACGGCAASAAGGTSMSSGT |
| | | nosZ2R | CAKRTGCAKSGCRTGGCAGAA |
| Phosphorus cycle | *phoC* | phoc-A-F1 | CGGCTCCTATCCGTCCGG |
| | | phoc-A-R1 | CAACATCGCTTTGCCAGTG |
| | *phoD* | ALPS-F730 | CAGTGGGACGACCACGAGGT |
| | | ALPS-R1101 | GAGGCCGATCGGCATGTCG |
| | *BPP* | bpp-F | GACGCAGCCGAYGAYCCNGCNITNTGG |
| | | bpp-R | CAGGSCGCANRTCIACRTTRTT |
| | *pqqC* | Fw | AACCGCTTCTACTACCAG |
| | | Rv | GCGAACAGCTCGGTCAG |
| Bacteria | 16S rRNA | 338F | ACTCCTACGGAGCGCA |
| | | 806R | GGACTACHVGGGTWTCTAAT |
| Fungi | ITS | ITS1F | CTTGGTCATTTAGAGGAAGTAA |
| | | ITS2R | GCTGCGTTCTTCATCGATGC |

**Table A4** Statistical table of bacterial and fungi species in both 0−10 cm and 10−20 cm soil layers in PPs and MPs.

| Microbial type | Soil layer (cm) | Stand type | Phylum | Class | Order | Family | Genus | OTU |
|---|---|---|---|---|---|---|---|---|
| Bacteria | 0−10 | PPs | 20 | 50 | 112 | 155 | 229 | 1435 |
| | | MPs | 21 | 62 | 131 | 187 | 283 | 1760 |
| | 10−20 | PPs | 20 | 47 | 108 | 155 | 224 | 1315 |
| | | MPs | 20 | 58 | 126 | 179 | 268 | 1695 |
| | Total | — | 21 | 64 | 140 | 201 | 311 | 1869 |
| Fungi | 0−10 | PPs | 8 | 18 | 41 | 57 | 73 | 693 |
| | | MPs | 8 | 21 | 45 | 73 | 93 | 723 |
| | 10−20 | PPs | 8 | 18 | 41 | 52 | 56 | 651 |
| | | MPs | 8 | 19 | 43 | 64 | 87 | 654 |
| | Total | — | 8 | 24 | 62 | 104 | 157 | 1128 |

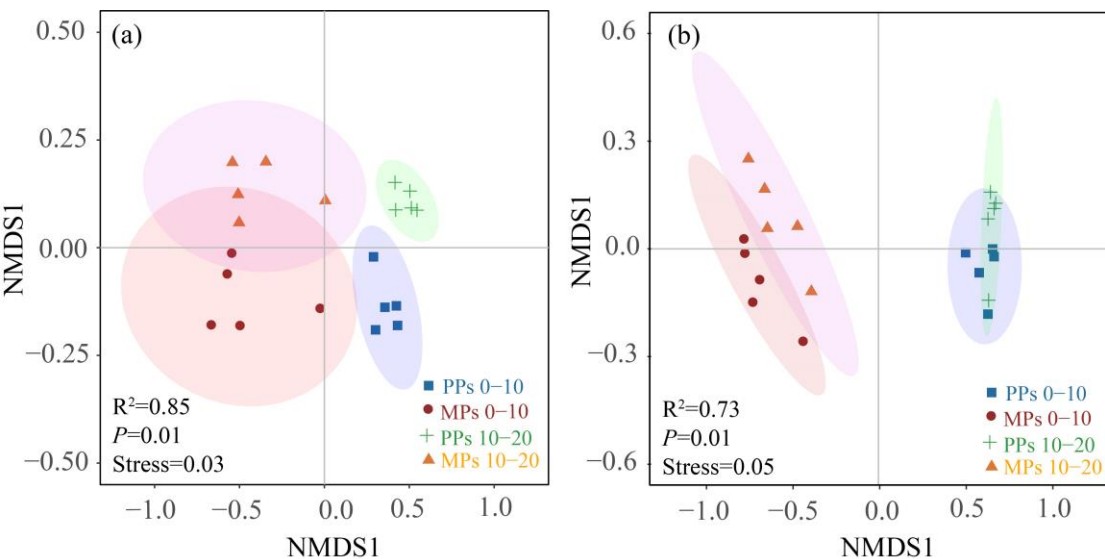

**Fig. A1** Nonmetric multidimensional scaling analysis of (a) bacterial and (b) fungal, based on Bray-Curtis similarity in both 0−10 cm and 10−20 cm soil layers in PPs and MPs.

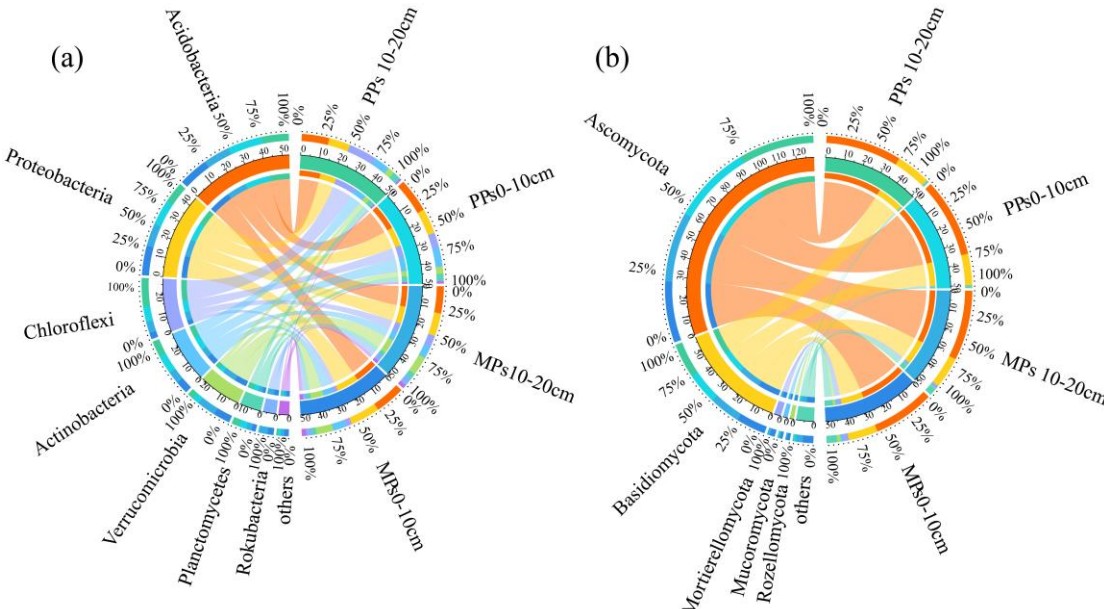

**Fig. A2** Chord diagrams showing the bacterial (a) and fungal (b) community composition (at the relative abundance >1% phylum level). The outer circle scale represents the percentage information of relative abundance of OTU in the sample; The inner circle scale represents the absolute abundance information of OTU in the sample (unit: 1000).

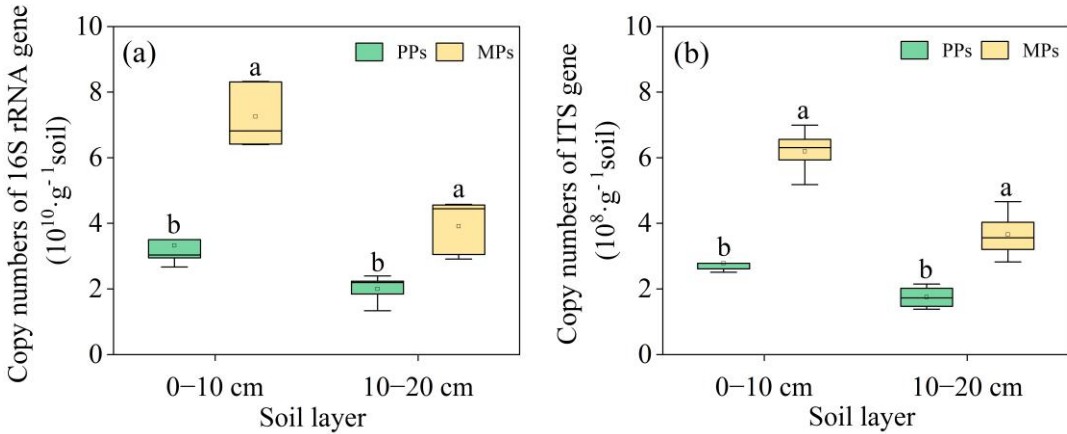

**Fig. A3** Comparisions copy number of (a) 16SrRNA and (b) ITS in both 0−10 cm and 10−20 cm soil layers in PPs and MPs.

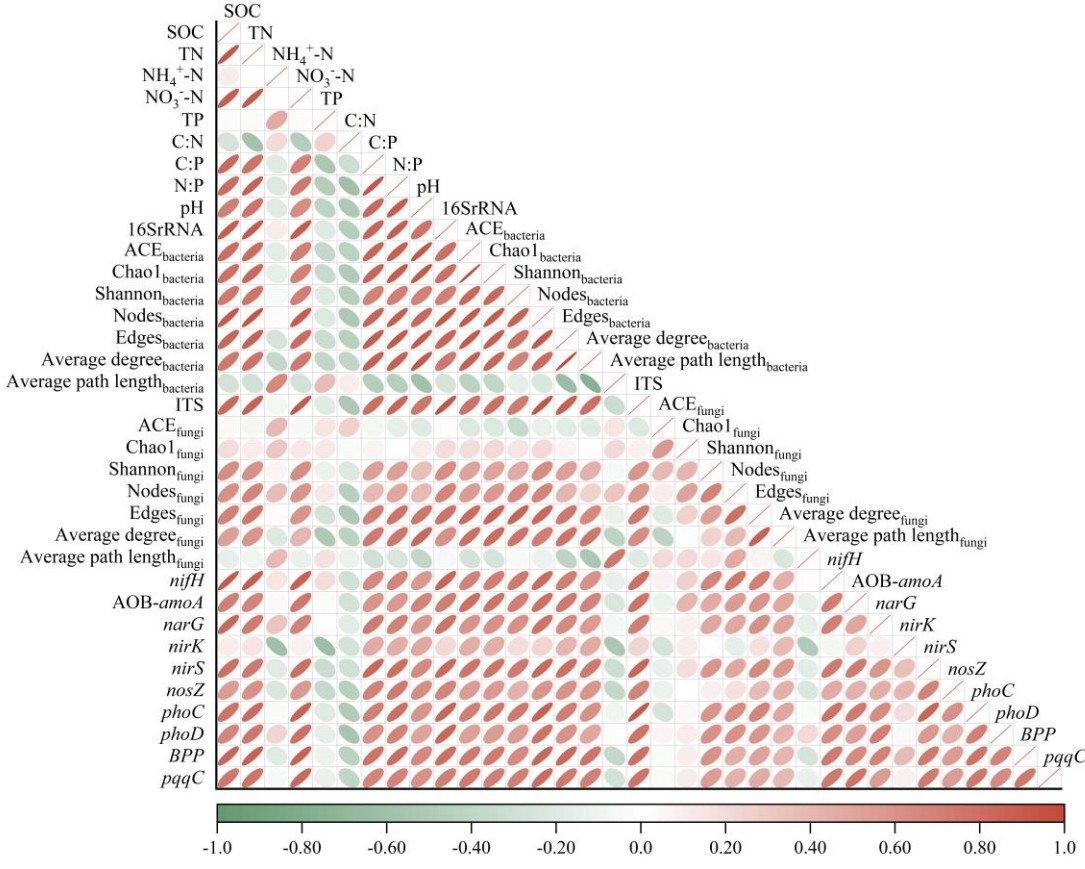

**Fig. A4** Correlative relationships between soil physico-chemical properties, microbial diversity
and complexity, and soil physico-chemical properties.
**Data availability**

5         All raw data can be provided by the corresponding authors upon request.

**Author contributions**
JL, XH, and YY conceived and designed of the study. JL, XH, YY, and WZ
processed and analyzed data acquisition of field experiments. JL, WZ, YL, HH, HM,
and QH conducted the fieldwork. JL and WZ performed laboratory analysis. JL
completed the analysis of the data and prepared the original draft of the manuscript,
XH, YY, YW, and AM helped to review and edit the manuscript. All the authors gave
approval for the final manuscript.

1  **Competing interests**

2      The authors declare that they have no conflict of interest.

**Acknowledgments**
This research was funded by grants from the National Natural Science
Foundation of China (Nos. 32171755, 32101500, and 31960240), the Guangxi
Natural Science Foundation (No. 2025GXNSFAA069288), and the scientific research
capacity building project for Youyiguan Forest Ecosystem Observation and Research
Station of Guangxi under Grant (No. 2203513003).

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
