# Peer review of "Soil microbial diversity and network complexity promote phosphorus"

_EGUsphere, 2024_

## Author Comment (AC1)

**Reviewer 1:** Jiyin Li and co-authors investigated the effects of monoculture plantation or mixed N-fixing tree species on soil phosphorus transformation in a eucalyptus plantation soil, they further tried to find explanatory biotic and abiotic factors for changes in transformation of phosphorus. I find this research potentially could contribute to our understanding of long-term (seventeen years) eucalyptus plantation soil restoration via introducing N-fixing tree species, particularly the interaction between two most important soil elements, i.e., N and P and related microbial diversity and network complexity. Overall, the topic was suitable for *Biogeosciences*, and would give some new points to the huge amount of soil nutrient restoration studies. But before it could be accepted for publishing, I have some questions and suggestions on the manuscript. First is a more detailed introduction on the experimental design, it would be ideal for providing detailed aspects of management (e.g., fertilizer amount and frequency, pesticide use) of the experimental plots. Second, I found the Materials and Methods was lack of relevant references. The authors should provide more references about the measuring methods of Soil properties and soil enzyme activity. Third, the organization and language of the paper need still need further modification. For example, in the discussion part, some contents belong to the repetition of the results, and the related and cited results can be summarized as supporting evidence without listing too many specific values (e.g., L404-406, L426-427, L487-490).

Response: We greatly appreciate your comments and suggestions, which are valuable in improving the quality of our manuscript and we will make necessary modification throughout the text. We believe the revision is much improved as a result of our modifications.

**Specific comments:**

1. L-26-L28: Please reorganize the sentence.

Response: Thank you for your suggestion. We have rephrased the sentence to avoid the confusion.

2. In this paper, your dissertation focuses on the introduction of N-fixing tree species to promote soil phosphorus transformation, but a description of the characteristics of N-fixing tree species is lacking in the introduction, please clarify.

Response: Thanks for pointing this out. More detail about the N-fixing tree species characteristics is provided in the Introduction section.

3. L-93: There is a notation error here.

Response: Corrected.

4. In the text, N-fixing tree should be changed to N-fixing tree species.

Response: Changed throughout the text.

5. Line 135 "soil extracellular enzymes" should use the abbreviation.

Response: Corrected.

6. In the text, you characterize the conversion of N and P in terms of soil enzyme activity, but you don't have a specific sentence in the text to describe it.

Response: Thanks for your comments. We indeed use acid phosphatase (ACP) to define P transformation, and we have added the expression that P transformation refers to ACP enzyme activity.

Relevant references are as follows:

[1] Nannipieri, P., Giagnoni, L., Landi, L., Renella, G.: Role of phosphatase enzymes in soil. Phosphorus in action: biological processes in soil phosphorus cycling, 215-243, https://doi.org/10.1007/978-3-642-15271-9_9, 2011.

[2] Yu, Q., Ma, S., Ni, X., Ni, X., Guo, Z., Tan, X., Zhong, M., Hanif, M. A., Zhu, J., Ji, C., Zhu, B., Fang, J.: Long-term phosphorus addition inhibits phosphorus transformations involved in soil arbuscular mycorrhizal fungi and acid phosphatase in two tropical rainforests. Geoderma, 425, 116076, https://doi.org/10.1016/j.geoderma.2022.11607,2022.

[3] Wang, Y., Luo, D., Xiong, Z., Wang, Z., Gao, M.: Changes in rhizosphere phosphorus fractions and phosphate-mineralizing microbial populations in acid soil as influenced by organic acid exudation. Soil Till. Res., 225, 105543, https://doi.org/10.1016/j.still.2022.105543, 2023.

7. L-93: Wrong colon space on this line.

Response: Corrected.

8. L262-263: You write "Significant ($P < 0.05$) increases in ……. were determined in both soil layers of the MPs and PPs ", I think it would be more precisely if you write "Significant ($P < 0.05$) higher of ……. were determined in both two investigated soil layers in MPs than those in PPs "

Response: Corrected.

9. L-374: Missing comma after "soil properties".

Response: Thanks for pointing this out. We have carefully checked and made necessary modification throughout the text.

10. L-439: Which specific result indicates that pH is the most crucial factor affecting microorganisms?

Response: Specified.

11. L441-L443: This sentence appears to have no correlation with the context and it is recommended to delete it.
Response: Deleted.

12. L-449-450: It is recommended to restructure the sentence to indicate that Proteobacteria encompasses Rhizobia.
Response: We have carefully checked the entire sentence and made appropriate.

13. L-454-455: The sentence is repetitive in meaning.
Response: Corrected.

14. The discussion indicated that these key microbial groups could increase the complexity of the network, but there were no corresponding results to support this view.
Response: Thanks for the comments. The corresponding results were shown in Fig. 4 and Fig. 5 in the Results section, and we will add some relevant descriptions in the Discussion section.

I highly value the large amount of work carried out by the authors. I hope my remarks will be
Response: Thanks for your good comments.

---

## Author Comment (AC2)

**Reviewer 2:** The authors examined key soil properties (e.g., pH, Nitrogen content, and Phosphorus content) and microbial diversity composition, comparing a *Eucalyptus* monoculture site with a *Eucalyptus-Acacia* mixed plantation site. They found significant differences in both soil chemical properties and microbial diversity composition between the two sites. This work and its topic align with the scope of *Biogeosciences*. However, the manuscript's organization and overall quality require significant improvement before it meets the publication standards of *Biogeosciences*.

Response: We greatly appreciate your comments and suggestions, which are valuable in improving the quality of our manuscript and we will make necessary modification throughout the text. We believe the revision is much improved as a result of our modifications.

**General comments:**

1. The key message emphasized throughout the manuscript—that "increased soil microbial diversity and network complexity has resulted in enhanced Phosphorus transformation" — appears to be overstated for several reasons:

While soil "P transformation" is repeatedly mentioned throughout the manuscript and included in data analyses (e.g., Figure 9), there is no clear definition or quantification methodology provided for this index. Phosphorus exists in soil in both organic and inorganic forms and undergoes continuous transformations through chemical, physical, and biological processes, making its transformation quantification complex. The authors only measured two P-related soil properties—total P (TP) and available P (AP), with AP notably absent from many follow-up analyses (Figure 8). It is unclear how the authors quantified "P-transformation" (which appears to be simply the Total P content) and drew conclusions about factors influencing this process.

Response: Thanks for pointing this out. We have checked the entire text and read numerous relevant references carefully. Phosphomonoesterase (i.e., ACP) mineralization is an essential strategy for P transformation (Nannipieri et al., 2011; Yu et al., 2022; Wang et al., 2023), so we employed soil ACP activity to analyse the dynamics of P transformation. In addition, we added some detailed description of P transformation to avoid the confusion.

References:

[1] Nannipieri, P., Giagnoni, L., Landi, L., Renella, G.: Role of phosphatase enzymes in soil. Phosphorus in action: biological processes in soil phosphorus cycling, 215-243, https://doi.org/10.1007/978-3-642-15271-9_9, 2011.

[2] Yu, Q., Ma, S., Ni, X., Ni, X., Guo, Z., Tan, X., Zhong, M., Hanif, M. A., Zhu,

J., Ji, C., Zhu, B., Fang, J.: Long-term phosphorus addition inhibits phosphorus transformations involved in soil arbuscular mycorrhizal fungi and acid phosphatase in two tropical rainforests. Geoderma, 425, 116076, https://doi.org/10.1016/j.geoderma.2022.11607,2022.

[3] Wang, Y., Luo, D., Xiong, Z., Wang, Z., Gao, M.: Changes in rhizosphere phosphorus fractions and phosphate-mineralizing microbial populations in acid soil as influenced by organic acid exudation. Soil Till. Res., 225, 105543, https://doi.org/10.1016/j.still.2022.105543, 2023.

2. The authors appear to have conflated correlation with causation in their narrative. While the *Eucalyptus* plantations were established in 2004 (approximately 20 years ago), soil sampling and analysis were conducted in 2021, providing only a recent snapshot. While it is reasonable to describe the observed differences in key soil chemical properties and microbial diversity between the two types of *Eucalyptus* plantations, the causal claim that increased soil microbial diversity and network complexity resulted in enhanced Phosphorus transformation is not adequately supported by the presented data and results.

Response: Thanks for your constructive feedback. The objective of our study is to investigate the mechanisms of microbial influence on phosphorus transformation in pure *Eucalyptus* plantations and mixed plantations of *Eucalyptus* and N-fixing trees species. So we think our observational data are convincing enough. Furthermore, in the future, we will continue to conduct relevant research.

3. Furthermore, according to the correlation matrix plot presented in Figure 8, Total P (TP) shows no significant correlations with either soil chemical properties or soil microbial diversity indices for most of the variable pairs. This lack of correlation directly contradicts the authors' main argument about the relationship between microbial diversity and Phosphorus transformation.

Response: Thanks for pointing this out. We apologize for confusing the reviewer. In the original manuscript we employed soil ACP activity to analyse the dynamics of P transformation. In addition, we will add some detailed description of P transformation to avoid the confusion.

4. The "Introduction"section requires substantial revision. It contains excessive methodological descriptions, such as Microbiome co-occurrence networks analysis and Functional gene markers, while lacking crucial discussions of key questions, mechanisms, patterns, and processes. Methodology merely describes the work conducted rather than establishing research significance. The interesting aspects that should be emphasized include the relationship

between N and P, the role of N-fixing plants in P transformation, the key players involved in these processes, and the main processes and influencing factors. Once these processes and key issues are clearly articulated, the methodological details would naturally fit into the Materials and Methods section. While the final paragraph includes hypotheses, these would be better integrated into the earlier parts of the Introduction.

Response: Thank you for your valuable suggestion. We will carefully check the entire Introduction section again and make appropriate.

5. The Introduction should address whether findings from *Eucalyptus* plantations can be generalized to other plantation types globally. Given the wide variety of both monoculture and mixed-species plantations worldwide, the authors should discuss how their research on *Eucalyptus* plantations relates to or differs from other plantation systems, and clarify the broader applicability of their findings.

Response: Thanks for good suggestions. We will improve the Introduction section based on your suggestions.

6. All expressions of "significant ($P < 0.05$)"should be revised to include the appropriate test statistics. Throughout the manuscript, the authors need to add the corresponding test statistics (g., t or F values) alongside the P-values to comply with standard statistical reporting conventions. For t-tests, results should be reported as ($t = XX, P < 0.05$), and for ANOVA tests, results should be reported as ($F = XX, P < 0.05$).

Response: We have revised it as suggested.

7. The manuscript contains numerous formatting errors in English text and symbols. For example:L2, Hyphens in title require spaces on both sides (e.g., "word - word" instead of "word-word"); redundant punctuation marks (e.g., double commas in L93); improper spacing in ratios (e.g., "C:N ratio" and "N:P ratio" should not have spaces around the colon); inconsistent hyphenation and capitalization in statistical terms (e.g., "z score" and "c score" should be "Z-score" and "C-score"). The authors should carefully review and correct all formatting issues throughout the manuscript, paying particular attention to: (1) proper use of hyphens and spaces; consistent capitalization; standard formatting of statistical terms; correct punctuation; proper ratio expressions

Response: We will carefully check the entire manuscript and make appropriate about the organization and language of the content to make it more readable.

**Specific comments:**

1. L48-51, The opening statement about Phosphorus being an essential nutrient is too absolute and lacks proper context.

Response: Thanks for your constructive suggestions. We have added some relevant contents to enrich the paragraph and made it appropriate.

2. L79-81, The statement "... is crucial for developing forest management strategies aimed at enhancing soil fertility and optimizing ecosystem functionality" is an overreaching conclusion that lacks sufficient support. In particular, the concept of ecosystem functionality was never a focus of this study.

Response: Thanks for your insightful comment. We have checked the sentence carefully and made necessary modification to avoid the confusion.

3. L236-237, the rationale for choosing these specific metrics (ACE, Chao1, and Shannon indices here in this study) over other available diversity measures for microbial community analysis should be explained.

Response: Thank you for your comment. We have read numerous relevant references carefully. Chao 1 and ACE indexes were used to estimate the richness of the bacterial and fungal community, while Shannon index was used to evaluate the diversity of bacterial and fungal community (Wang et al., 2018; Sun et al.,2021; Qiu et al., 2021; Malard et al., 2022). Therefore, these indices combined provide a more reliable and comprehensive view of microbial community structure and its potential links to soil nutrient cycling.

Relevant references are as follows:

Wang, C., Liu, D., Bai, E.: Decreasing soil microbial diversity is associated with decreasing microbial biomass under nitrogen addition. Soil Biol. Biochem., 120, 126-133, https://doi.org/10.1016/j.soilbio.2018.02.003, 2018.

Sun, Y., Ren, X., Rene, E. R., Wang, Z., Zhou, L., Zhang, Z., Wang, Q.: The degradation performance of different microplastics and their effect on microbial community during composting process. Bioresource Technol., 332, 125133, https://doi.org/10.1016/j.biortech.2021.12513, 2021.

Qiu, L., Zhang, Q., Zhu, H., Reich, P. B., Banerjee, S., van der Heijden, M. G., Sadowsky M. J., Ishii S., Jia X., Shao M., Liu B., Jiao H., Li H., Wei, X.: Erosion reduces soil microbial diversity, network complexity and multifunctionality. ISME J., 15(8), 2474-2489, https://doi.org/10.1038/s41396-021-00913-1, 2021.

Malard, L. A., Mod, H. K., Guex, N., Broennimann, O., Yashiro, E., Lara, E., Mitchell, A. D. E., Niculita-Hirzel, H., Guisan, A.: Comparative analysis of diversity and environmental niches of soil bacterial, archaeal, fungal and protist communities reveal niche divergences along environmental gradients in the Alps. Soil Biol. Biochem., 169, 108674, https://doi.org/10.1016/j.soilbio.2022.108674, 2022.

4. L262-263, The description of results is unclear regarding which group showed an increase when compared to which group.

Response: Thank you for your suggestion. We have rephrased the sentence to avoid the confusion

"Significant ($P < 0.05$) higher of SOC, TN, $NO_3^-$-N, C: P, N: P, and pH were determined in both two investigated soil layers in MPs than those in PPs (Table 1)."

5. L245, There are inconsistent statements about the correlation analysis method used: L245 mentions Pearson correlation, L362 refers to Spearman correlation analysis, and Fig. 8 (L372) again states Pearson correlations. The authors need to clarify which correlation method was actually used and maintain consistency throughout the manuscript.

Response: Thanks for your careful checks. We used Pearson's correlation analysis and made appropriate modified to make the word harmonized throughout the text.

6. L372, Figure 8's readability is poor due to the excessive number of correlated variables. With many variables showing covariation, it is difficult to identify meaningful relationships. The authors should justify the purpose of including so many variables in the correlation analysis and consider focusing on key variables that address their research questions.

Response: Phosphorus transformation is directly or indirectly influenced by a variety of biotic and abiotic factors, and there exist unknown interactions among the factors. Therefore, we need to systematically explore the interactions among the factors in order to support the subsequent discussions.

For example: "In line with our results, a higher abundance and diversity of *phoD-*, *phoC-*, and *pqqC*-bearing soil microorganisms; higher abundances of these genes in soil were correlated with higher soil SOC and TN contents (Fig. 8) (Luo et al., 2019; Cao et al., 2022). Our study also identified significantly positive correlations between most N and P functional genes and 16S rRNA as well as the ACE, Chao1, and Shannon indexes in bacterial communities, whereas a significant positive correlation was determined only between the ITS region and the Shannon index in fungal communities (Fig. 8)."

7. L389-393, The meaning and purpose of Figure 9 are unclear. The figure caption only describes the visual elements but lacks explanation of what the figure aims to demonstrate or illustrate.

Response: Thanks for pointing this out. We added some detailed descriptions and made it clear and specific.

"Fig. 9a shows the directed graph of the partial least squares path models (PLS-PM), and Fig. 9b shows the Standardized total effects (direct plus indirect effects) on P transformation derived from the PLS-PM."

8. L406-408 There is a logical inconsistency in the manuscript's core arguments. While L406-408 emphasizes how soil properties influence microbial community composition ("Soil properties are key in influencing the composition of microbial communities..."), the main thesis appears to argue that differences in microbial community diversity lead to variations in soil P transformation.

Response: Thanks for pointing these out. We have carefully checked the entire manuscript again and will make necessary modification to avoid the confusion.

9. L465-468, the text here is redundant as similar sentences appear in the Introduction. Moreover, this background information belongs in the Introduction section rather than the Discussion, where the focus should be on interpreting results and their implications.

Response: Thank you for your feedback. We will reorganize the Discussion section and expand our discussion which correspond tightly to the hypotheses and core findings of our study.

10. L508-L512 This is for sure. The introduced trees are N-fixing trees.

Response: Thank you for your suggestion. We have rephrased the sentence to avoid the confusion.

---

## Author Comment (AC3)

**Reviewer 3:** This study quantified soil fungal and bacterial communities, genes, and networks for both pure *Eucalyptus* (PP) and mixed *Eucalyptus-Acacia* (MP) plantations. The plantations have been growing for 17 years, allowing authors to report long-term differences caused by co-planting *Eucalyptus* with a nitrogen-fixing tree species. The results are interesting, consisting of many differences between the plantation types in the composition and function of the microbial communities. Although I cannot address many of molecular methods, as they are outside of the scope of my expertise, I hope my comments below help improve the manuscript. Once they are addressed, I believe it will be a good fit for *Biogeosciences*.

Response: Thanks for your good comments.

1. The hypotheses presented in the last paragraph of the introduction are unclear. For (1), it is stated that diversity and composition of soil microorganisms will change with mixed planting. How will they change? For (2), "mixed plantations intensify the response to the beneficial impacts of N-fixing tree" is unclear and should be reworded. For (3), this hypothesis seems to overlap with hypothesis (1) (both mention diversity), but is more specific, suggesting that there will be higher diversity in mixed plantations.

Response: As suggested, we rephrased the hypotheses, and made modification accordingly.

"We proposed the following hypotheses: (1) tree species mixing would alter the composition of soil microbial communities and increase microbial diversity and network complexity in the soil, and (2) the soil P transformation driven by tree species mixing may be positively regulated by microbial diversity and network complexity."

The rationale for making measurements at the two depths (0-10 and 10-20cm) are unclear. Please provide an explanation for why these two depths were chosen.

Response: Thanks for your comment. According to our previous soil investigation, collecting soil samples from two layers can more systematically and comprehensively explore the influence mechanism of different factors on soil phosphorus conversion. Furthermore, this approach ensures that the resultant observational datasets exhibit enhanced representativeness by minimising vertical heterogeneity artefacts inherent to single-layer sampling protocols.

2. The rationale for the different alpha index analyses (ACE, Chao1, Shannon) should be mentioned. That is, why are all three used and in what ways do insights from them differ?

Response: Thank you for your comment. We have read numerous relevant references carefully. Chao 1 and ACE indexes were used to estimate the richness of the bacterial and fungal community, while Shannon index was

used to evaluate the diversity of bacterial and fungal community (Wang et al., 2018; Sun et al.,2021; Qiu et al., 2021; Malard et al., 2022). Therefore, these indices combined provide a more reliable and comprehensive view of microbial community structure and its potential links to soil nutrient cycling.

Relevant references are as follows:

Sun, Y., Ren, X., Rene, E. R., Wang, Z., Zhou, L., Zhang, Z., Wang, Q.: The degradation performance of different microplastics and their effect on microbial community during composting process. Bioresource Technol., 332, 125133, https://doi.org/10.1016/j.biortech.2021.12513, 2021.

Qiu, L., Zhang, Q., Zhu, H., Reich, P. B., Banerjee, S., van der Heijden, M. G., Sadowsky M. J., Ishii S., Jia X., Shao M., Liu B., Jiao H., Li H., Wei, X.: Erosion reduces soil microbial diversity, network complexity and multifunctionality. ISME J., 15(8), 2474-2489, https://doi.org/10.1038/s41396-021-00913-1, 2021.

Malard, L. A., Mod, H. K., Guex, N., Broennimann, O., Yashiro, E., Lara, E., Mitchell, A. D. E., Niculita-Hirzel, H., Guisan, A.: Comparative analysis of diversity and environmental niches of soil bacterial, archaeal, fungal and protist communities reveal niche divergences along environmental gradients in the Alps. Soil Biol. Biochem., 169, 108674, https://doi.org/10.1016/j.soilbio.2022.108674, 2022.

Wang, C., Liu, D., Bai, E.: Decreasing soil microbial diversity is associated with decreasing microbial biomass under nitrogen addition. Soil Biol. Biochem., 120, 126-133, https://doi.org/10.1016/j.soilbio.2018.02.003, 2018.

3. It would be helpful to mention the perceived function of the different genes that were measured. For example, in the paragraph at L198 and in Figs. 5-7.

Response: Thank you for your suggestion. We have added some details about perceived function of the different genes in the methods and Figs. 5-7 sections to make it more readable.

4. I think that there should be a discussion of why there was higher TP in PPs than MPs and whether trees in MPs and PPs might differ in whether they are limited by N vs. P.

Response: Thanks for pointing this out. Detailed descriptions were added in the Discussion section.

5. The introduction and discussion would benefit from discussing mixed plantations between N-fixing and non-fixing trees in general. How representative are *Eucalyptus-Acacia* plantations of mixed plantations elsewhere?

Response: Thank you for your valuable suggestion. We have carefully re-checked the Introduction and Discussion sections and will add more relevant content of mixed plantations between N-fixing and non-fixing trees. In addition, we will add relevant supporting references about *Eucalyptus-Acacia* plantations of mixed plantations.

6. The direction of causality is unclear. Throughout the manuscript, the authors argue that microbial diversity, structure, complexity promote P transformation. However, sentences such as that on L68-70 suggest causality is in the other direction.

Response: Thank you for your suggestion. We have rephrased the sentence to avoid the confusion.

7. The manuscript should be checked for typos and grammar. There are many instances of minor mistakes.

Response: We will carefully check the entire manuscript and make appropriate about the organization and language of the content to make it more readable.

**Specific comments:**

1. Title: I would change to: "Soil microbial diversity and network complexity promote phosphorus transformation: A case of long-term mixed-species plantations of *Eucalyptus* with a nitrogen-fixing tree species"

Response: Changed.

2. L24-26: Clarify that the study was in both PPs and MPs. The sentence makes it sound like the study was just done in PPs.

Response: Corrected as follows:

"Therefore, we conducted a 17-year field experiment in pure *Eucalyptus* plantations (PPs) and mixed plantations (MPs) of *Eucalyptus* and N-fixing trees species to assess the effects of soil P transformation, with data collected from two soil layers: 0-10 cm and 10-20 cm depths."

3. L30: The two soil layers tested should probably be mentioned before reporting specific results for one of them.

Response: Specified.

4. L63: "soil health" is a vague statement. Be more specific.

Response: Specified.

5. L95: This sentence states that N content influences soil pH. Typically, the direction is one where an increase in N content lowers soil pH. The results show that pH however increased, which I found surprising. Although the discussion has a few lines on why, it may be good to address the hypothesized direction of change somewhere in the introduction.

Response: Thank you for your valuable suggestion. We will add some detailed description and make necessary modification.

6. L99: Change the part of the sentence that follows the comma to "thereby accelerating nutrient cycling and improving soil fertility"

Response: Corrected.

7. L106: It is unclear what is meant by "soil nutrient effectiveness".

Response: Thanks for pointing this out. We have rephrased the sentence to avoid the confusion.

"However, monocultures and short-term rotation management of *Eucalyptus* plantation have led to soil degradation, reductions in soil nutrient effectiveness (i.e., the availability of nutrients such as nitrogen, phosphorus, and potassium in forms that can be absorbed and utilized by plants), and soil microbial function and diversity, as well as other adverse ecological effects."

8. L111: Replace "fewer" with "less or no"

Response: Changed.

9. L117: This might be a good time to mention the N-fixing tree species that is used in the MPs.

Response: We have revised it as suggested.

10. L125-126: I am unsure of what is meant by "along with genes associated with N and P cycling".

Response: Accepted and it has been revised in the manuscript. Now read like: " along with genes involved in N and P cycling processes, regulate P transformation".

11. L262-263: Clarify that the increase was in going from PPs to MPs.

Response: Corrected.

12. L305: Can you explain by what metric pH is the most important regulator? It is not immediately clear from looking at Figure 3b.

Response: Thank you for your comment. The soil physicochemical properties influencing the variations of dominant microorganism phyla were identified by using redundancy analysis (RDA). The sequential selection process of RDA was used to identify the drastically distinguishing variables for soil physicochemical properties and specific microorganism phyla. Significant variables ($P < 0.05$) were employed in subsequent analysis. In our study, the value of pH ($F = 4.3$, $P = 0.003$) had the greatest impact compared to other factors ($P > 0.05$).

13. L376: Please provide a number for the "high goodness of fit."
Response: Added.

14. L450-451: Having actinobacteria in this sentence is misleading. Actinorhizal plants form N-fixing symbioses with Frankia, which are actinobacteria. However, Acacia is not an actinorhizal N fixer. Instead, Acacia forms N-fixing symbioses with Rhizobia, which are Proteobacteria.
Response: Thank you for your suggestion. We have rephrased the sentence to avoid the confusion

15. Table 1: Clarify whether the +/- refers to the standard deviation or the standard error.
Response: Clarified.

16. Table 2: Bacteria is misspelled.
Response: Corrected.

17. Figure 1: In the caption mention the threshold p value (my guess is p < 0.05) that determines whether differences between treatments are significant or not.
Response: Added.

18. Figure 4: The Zi-Pi plots have the connectors (high among module connectivity) and module hubs (high within module connectivity) switched in the legend. Also, it is not clear what is meant by "node color node size" in the caption.
Response: Corrected.

19. Figure 9: The caption appears to explain 9a, but not 9b.
Response: Thanks for pointing this out. We added some detailed descriptions and made it clear and specific.
"Figure 9b presents the Standardized total effects (direct plus indirect effects) on P transformation derived from the PLS-PM."

---

## Author Response (AR1)

Dear Prof. Goll and Reviewers,

Thank you for your kind consideration of our manuscript "Promoted phosphorus transformation by increasing soil microbial diversity and network complexity - A case of long-term mixed-species plantations of *Eucalyptus* with N-fixing tree species (Manuscript Number: egusphere-2024-3456)" for publication in *Biogeosciences*. We are grateful for the opportunity to share our research with the journal's readership and for the constructive comments from this issue's editor and the three anonymous reviewers. Responding to specific editor and reviewer comments, we have revised the manuscript for clarity and added the information requested. We provide specific responses to editor and reviewer comments below.

Editor comments:

thank you very much for the response to the reviewer comments. I would like to ask you to pay attention on RC2 concern #2 regarding causality, your response is not addressing the argument of the reviewer. You need to provide arguments which support the direction of causality you proposed or adjust the interpretation of results.

Response: Thanks for the opportunity to revise our work. We have carefully re-checked the entire manuscript and made corrections according to the Reviewers' comments.

The reviewers agree about deficiencies in presentation of results (structure, gaps in methods, etc). Please pay attention to address them. Figure 8 is indeed quite complex & only a subset of relationships are discussed in the manuscript, consider adding a more focused figure 8 and move the current form to SI.

Response: Thanks for your positive comments and a nice summary of our work. We have made corrections according to the Reviewers' comments. We have added a more focused and concise Figure 8 and moved the current form to Figure S4 in the supplementary material. Our responses are listed below.

Reviewer comments:

**Reviewer 1:** Jiyin Li and co-authors investigated the effects of monoculture plantation or mixed N-fixing tree species on soil phosphorus transformation in a eucalyptus plantation soil, they further tried to find explanatory biotic and abiotic factors for changes in transformation of phosphorus. I find this research potentially could contribute to our understanding of long-term (seventeen years) eucalyptus plantation soil restoration via introducing N-fixing tree species, particularly the interaction between two most important soil elements, i.e., N and P and related microbial diversity and network complexity. Overall, the topic was suitable for *Biogeosciences*, and would give some new points to the huge amount of soil nutrient restoration studies. But

before it could be accepted for publishing, I have some questions and suggestions on the manuscript. First is a more detailed introduction on the experimental design, it would be ideal for providing detailed aspects of management (e.g., fertilizer amount and frequency, pesticide use) of the experimental plots. Second, I found the Materials and Methods was lack of relevant references. The authors should provide more references about the measuring methods of Soil properties and soil enzyme activity. Third, the organization and language of the paper need still need further modification. For example, in the discussion part, some contents belong to the repetition of the results, and the related and cited results can be summarized as supporting evidence without listing too many specific values (e.g., L404-406, L426-427, L487-490).

Response: We greatly appreciate your comments and suggestions. We greatly appreciate your comments and suggestions. A more detailed introduction of the experimental design has been provided in the "Plot design and sampling" section (L162-165). Additionally, we have added more references in the "Materials and Methods" section (L191-192,L193,L198), and the result descriptions in the "Discussion" have been summarized (L456-458).

**Specific comments:**

1. L-26-L28: Please reorganize the sentence.

Response: Reorganized (L28-30).

2. In this paper, your dissertation focuses on the introduction of N-fixing tree species to promote soil phosphorus transformation, but a description of the characteristics of N-fixing tree species is lacking in the introduction, please clarify.

Response: Thanks for pointing this out. More detail about the N-fixing tree species characteristics is provided in the Introduction section (L118-127).

3. L-93: There is a notation error here.

Response: Corrected.

4. In the text, N-fixing tree should be changed to N-fixing tree species.

Response: Changed throughout the text.

5. Line 183 "soil extracellular enzymes" should use the abbreviation.

Response: Corrected (L206).

6. In the text, you characterize the conversion of N and P in terms of soil enzyme activity, but you don't have a specific sentence in the text to describe it.

Response: Thanks for pointing this out. We have checked the entire text and read numerous relevant references carefully. Phosphomonoesterase (i.e., ACP) mineralization is an essential strategy for P transformation (Luo et al., 2019; Yu et al., 2022; Wang et al., 2023), so we employed soil ACP activity to analyse the dynamics of P transformation. In addition, we added some detailed description of P transformation in the Introduction (L132-135) to avoid the confusion.

References:

[1] Luo G, Sun B, Li L, Li M, Liu M, Zhu Y, Guo S, Ling N, Shen Q. Understanding how long-term organic amendments increase soil phosphatase activities: insight into phoD-and phoC-harboring functional microbial populations. Soil Biology and Biochemistry. 2019, 139: 107632.

[2] Yu Q, Ma S, Ni X, Ni X, Guo Z, Tan X, Zhong M, Hanif MA, Zhu J, Ji C, Zhu B. Long-term phosphorus addition inhibits phosphorus transformations involved in soil arbuscular mycorrhizal fungi and acid phosphatase in two tropical rainforests. Geoderma. 2022, 425: 116076.

[3] Wang Y, Luo D, Xiong Z, Wang Z, Gao M. Changes in rhizosphere phosphorus fractions and phosphate-mineralizing microbial populations in acid soil as influenced by organic acid exudation. Soil and Tillage Research. 2023, 225: 105543.

7. L-93: Wrong colon space on this line.

Response: Corrected.

8. L262-263: You write "Significant ($P < 0.05$) increases in ……. were determined in both soil layers of the MPs and PPs ", I think it would be more precisely if you write "Significant ($P < 0.05$) higher of ……. were determined in both two investigated soil layers in MPs than those in PPs "

Response: Corrected (L290-291).

9. L-374: Missing comma after "soil properties".

Response: Corrected.

10. L-439: Which specific result indicates that pH is the most crucial factor affecting microorganisms?

Response: Specified (L507).

11. L441-L443: This sentence appears to have no correlation with the context and it is

recommended to delete it.

Response: Deleted.

12. L-449-450: It is recommended to restructure the sentence to indicate that Proteobacteria encompasses Rhizobia.

Response: We have carefully checked the entire sentence and made appropriate (L515-516).

13. L-454-455: The sentence is repetitive in meaning.

Response: Corrected (L520-521).

14. The discussion indicated that these key microbial groups could increase the complexity of the network, but there were no corresponding results to support this view.

Response: Thanks for the comments. The corresponding results were shown in Fig. 4 in the Results section (L363).

I highly value the large amount of work carried out by the authors. I hope my remarks will be valuable for the authors.

Response: Thanks for your good comments.

**Reviewer 2:** The authors examined key soil properties (e.g., pH, Nitrogen content, and Phosphorus content) and microbial diversity composition, comparing a *Eucalyptus* monoculture site with a *Eucalyptus-Acacia* mixed plantation site. They found significant differences in both soil chemical properties and microbial diversity composition between the two sites. This work and its topic align with the scope of *Biogeosciences*. However, the manuscript's organization and overall quality require significant improvement before it meets the publication standards of *Biogeosciences*.

Response: We greatly appreciate your comments and suggestions, which are valuable in improving the quality of our manuscript. We believe the revision is much improved as a result of our modifications.

**General comments:**

1. The key message emphasized throughout the manuscript—that "increased soil microbial diversity and network complexity has resulted in enhanced Phosphorus transformation" — appears to be overstated for several reasons:

While soil "P transformation" is repeatedly mentioned throughout the manuscript and included in data analyses (e.g., Figure 9), there is no clear definition or quantification methodology provided for this index. Phosphorus exists in soil in both organic and inorganic forms and undergoes continuous transformations through chemical, physical, and biological processes, making its transformation quantification complex. The authors only measured two P-related soil properties—total P (TP) and available P (AP), with AP notably absent from many follow-up analyses (Figure 8). It is unclear how the authors quantified "P-transformation" (which appears to be simply the Total P content) and drew conclusions about factors influencing this process.

Response: Thanks for pointing this out. We have checked the entire text and read numerous relevant references carefully. Phosphomonoesterase (i.e., ACP) mineralization is an essential strategy for P transformation (Luo et al., 2019; Yu et al., 2022; Wang et al., 2023), so we employed soil ACP activity to analyse the dynamics of P transformation. In addition, we added some detailed description of P transformation in the Introduction (L132-135) to avoid the confusion.

2. The authors appear to have conflated correlation with causation in their narrative. While the *Eucalyptus* plantations were established in 2004 (approximately 20 years ago), soil sampling and analysis were conducted in 2021, providing only a recent snapshot. While it is reasonable to describe the observed differences in key soil chemical properties and microbial diversity between the two types of *Eucalyptus* plantations, the causal claim that increased soil microbial diversity and network complexity resulted in enhanced Phosphorus transformation is not

adequately supported by the presented data and results.

Response: Thanks for your constructive feedback. The objective of our study is to investigate the mechanisms of microbial influence on phosphorus transformation in pure *Eucalyptus* plantations and mixed plantations of *Eucalyptus* and N-fixing trees species. So we think our observational data are convincing enough. Furthermore, in the future, we will continue to conduct relevant research.

3. Furthermore, according to the correlation matrix plot presented in Figure 8, Total P (TP) shows no significant correlations with either soil chemical properties or soil microbial diversity indices for most of the variable pairs. This lack of correlation directly contradicts the authors' main argument about the relationship between microbial diversity and Phosphorus transformation.

Response: Thanks for pointing this out. We apologize for confusing the reviewer. In the original manuscript we employed soil ACP activity to analyse the dynamics of P transformation. In addition, we have added some detailed description of P transformation to avoid the confusion (L132-135).

4. The "Introduction" section requires substantial revision. It contains excessive methodological descriptions, such as Microbiome co-occurrence networks analysis and Functional gene markers, while lacking crucial discussions of key questions, mechanisms, patterns, and processes. Methodology merely describes the work conducted rather than establishing research significance. The interesting aspects that should be emphasized include the relationship between N and P, the role of N-fixing plants in P transformation, the key players involved in these processes, and the main processes and influencing factors. Once these processes and key issues are clearly articulated, the methodological details would naturally fit into the Materials and Methods section. While the final paragraph includes hypotheses, these would be better integrated into the earlier parts of the Introduction.

Response: Thank you for your valuable suggestion. We have significantly revised the "Introduction" section, including discussions on key questions, mechanisms, patterns, and processes (L80-88, L97-100). We have also added detailed descriptions of the relationship between N and P, the role of N-fixing plants in P transformation, the key players involved in these processes, and the main processes and influencing factors (L51-54, L55-58, L68-75, L118-130). Additionally, the content of the hypotheses is now introduced earlier in the Introduction (L72-75, L84-88, L106-108).

5. The Introduction should address whether findings from *Eucalyptus* plantations can be

generalized to other plantation types globally. Given the wide variety of both monoculture and mixed-species plantations worldwide, the authors should discuss how their research on *Eucalyptus* plantations relates to or differs from other plantation systems, and clarify the broader applicability of their findings.

Response: Thanks for good suggestions. We have improved the Introduction section based on your suggestions (L118-130).

6. All expressions of "significant ($P < 0.05$)" should be revised to include the appropriate test statistics. Throughout the manuscript, the authors need to add the corresponding test statistics (g., t or F values) alongside the P-values to comply with standard statistical reporting conventions. For t-tests, results should be reported as (t = XX, $P < 0.05$), and for ANOVA tests, results should be reported as (F = XX, $P < 0.05$).

Response: We have revised it as suggested (Table 1, L301-304, etc.).

7. The manuscript contains numerous formatting errors in English text and symbols. For example:L2, Hyphens in title require spaces on both sides (e.g., "word - word" instead of "word-word"); redundant punctuation marks (e.g., double commas in L93); improper spacing in ratios (e.g., "C:N ratio" and "N:P ratio" should not have spaces around the colon); inconsistent hyphenation and capitalization in statistical terms (e.g., "z score" and "c score" should be "Z-score" and "C-score"). The authors should carefully review and correct all formatting issues throughout the manuscript, paying particular attention to: (1) proper use of hyphens and spaces; consistent capitalization; standard formatting of statistical terms; correct punctuation; proper ratio expressions

Response: Thanks for pointing this out. We have checked the entire manuscript carefully and made appropriate about the formatting of the text.

**Specific comments:**
1. L48-51, The opening statement about Phosphorus being an essential nutrient is too absolute and lacks proper context.

Response: Revised (L49-50).

2. L79-81, The statement "... is crucial for developing forest management strategies aimed at enhancing soil fertility and optimizing ecosystem functionality" is an overreaching conclusion that lacks sufficient support. In particular, the concept of ecosystem functionality was never a focus of this study.

Response: Thanks for your insightful comment. We have checked the sentence carefully and revised necessary to make appropriate (L86-88).

3. L236-237, the rationale for choosing these specific metrics (ACE, Chao1, and Shannon indices here in this study) over other available diversity measures for microbial community analysis should be explained.

Response: Thank you for your comment. We have read numerous relevant references carefully. Chao 1 and ACE indexes were used to estimate the richness of the bacterial and fungal community, while Shannon index was used to evaluate the diversity of bacterial and fungal community (Wang et al., 2018; Sun et al.,2021; Qiu et al., 2021; Malard et al., 2022). Therefore, these indices combined provide a more reliable and comprehensive view of microbial community structure and its potential links to soil nutrient cycling. In addition, we have added some detailed description about this in Data analyses (L259-261).

Relevant references are as follows:

Wang, C., Liu, D., Bai, E.: Decreasing soil microbial diversity is associated with decreasing microbial biomass under nitrogen addition. Soil Biol. Biochem., 120, 126-133, https://doi.org/10.1016/j.soilbio.2018.02.003, 2018.

Sun, Y., Ren, X., Rene, E. R., Wang, Z., Zhou, L., Zhang, Z., Wang, Q.: The degradation performance of different microplastics and their effect on microbial community during composting process. Bioresource Technol., 332, 125133, https://doi.org/10.1016/j.biortech.2021.12513, 2021.

Qiu, L., Zhang, Q., Zhu, H., Reich, P. B., Banerjee, S., van der Heijden, M. G., Sadowsky M. J., Ishii S., Jia X., Shao M., Liu B., Jiao H., Li H., Wei, X.: Erosion reduces soil microbial diversity, network complexity and multifunctionality. ISME J., 15(8), 2474-2489, https://doi.org/10.1038/s41396-021-00913-1, 2021.

Malard, L. A., Mod, H. K., Guex, N., Broennimann, O., Yashiro, E., Lara, E., Mitchell, A. D. E., Niculita-Hirzel, H., Guisan, A.: Comparative analysis of diversity and environmental niches of soil bacterial, archaeal, fungal and protist communities reveal niche divergences along environmental gradients in the Alps. Soil Biol. Biochem., 169, 108674, https://doi.org/10.1016/j.soilbio.2022.108674, 2022.

4. L262-263, The description of results is unclear regarding which group showed an increase when compared to which group.

Response: Corrected (L290-291).

5. L245, There are inconsistent statements about the correlation analysis method used: L245 mentions Pearson correlation, L362 refers to Spearman correlation analysis, and Fig. 8 (L372) again states Pearson correlations. The authors need to clarify which correlation method was

actually used and maintain consistency throughout the manuscript.

Response: Thanks for pointing this out. We used Pearson's correlation analysis and made appropriate modified (L238, L269, L423).

6. L372, Figure 8's readability is poor due to the excessive number of correlated variables. With many variables showing covariation, it is difficult to identify meaningful relationships. The authors should justify the purpose of including so many variables in the correlation analysis and consider focusing on key variables that address their research questions.

Response: Thank you for your suggestions. Phosphorus transformation is directly or indirectly influenced by a variety of biotic and abiotic factors, and there exist unknown interactions among the factors. Therefore, we need to systematically explore the interactions among the factors in order to support the subsequent discussions. We have moved the correlation heat map in the form of Fig. 8 to Fig. S4 and created a new figure for Fig. 8 (L420).

7. L389-393, The meaning and purpose of Figure 9 are unclear. The figure caption only describes the visual elements but lacks explanation of what the figure aims to demonstrate or illustrate.

Response: Thanks for pointing this out. We added some detailed descriptions and made it clear and specific (L440-446).

8. L406-408 There is a logical inconsistency in the manuscript's core arguments. While L406-408 emphasizes how soil properties influence microbial community composition ("Soil properties are key in influencing the composition of microbial communities..."), the main thesis appears to argue that differences in microbial community diversity lead to variations in soil P transformation.

Response: Thanks for pointing these out. We have carefully checked the entire manuscript again and have make necessary modification to avoid the confusion (L452-456).

9. L465-468, the text here is redundant as similar sentences appear in the Introduction. Moreover, this background information belongs in the Introduction section rather than the Discussion, where the focus should be on interpreting results and their implications.

Response: Deleted.

10. L508-L512 This is for sure. The introduced trees are N-fixing trees.

Response: Corrected (L571-575).

**Reviewer 3:** This study quantified soil fungal and bacterial communities, genes, and networks for both pure *Eucalyptus* (PP) and mixed *Eucalyptus-Acacia* (MP) plantations. The plantations have been growing for 17 years, allowing authors to report long-term differences caused by co-planting *Eucalyptus* with a nitrogen-fixing tree species. The results are interesting, consisting of many differences between the plantation types in the composition and function of the microbial communities. Although I cannot address many of molecular methods, as they are outside of the scope of my expertise, I hope my comments below help improve the manuscript. Once they are addressed, I believe it will be a good fit for *Biogeosciences*.

Response: Thanks for your good comments.

1. The hypotheses presented in the last paragraph of the introduction are unclear. For (1), it is stated that diversity and composition of soil microorganisms will change with mixed planting. How will they change? For (2), "mixed plantations intensify the response to the beneficial impacts of N-fixing tree" is unclear and should be reworded. For (3), this hypothesis seems to overlap with hypothesis (1) (both mention diversity), but is more specific, suggesting that there will be higher diversity in mixed plantations.

Response: As suggested, we rephrased the hypotheses, and made modification accordingly (L135-139).

The rationale for making measurements at the two depths (0-10 and 10-20cm) are unclear. Please provide an explanation for why these two depths were chosen.

Response: Thanks for your comment. According to our previous soil investigation, collecting soil samples from two layers can more systematically and comprehensively explore the influence mechanism of different factors on soil phosphorus conversion. Furthermore, this approach ensures that the resultant observational datasets exhibit enhanced representativeness by minimising vertical heterogeneity artefacts inherent to single-layer sampling protocols. we have added some detailed description about this in "2.2 Plot design and sampling"(L174-178).

2. The rationale for the different alpha index analyses (ACE, Chao1, Shannon) should be mentioned. That is, why are all three used and in what ways do insights from them differ?

Response: Thank you for your comment. We have read numerous relevant references carefully. Chao 1 and ACE indexes were used to estimate the richness of the bacterial and fungal community, while Shannon index was used to evaluate the diversity of bacterial and fungal community (Wang et al., 2018; Sun et al.,2021; Qiu et al., 2021; Malard et al., 2022). Therefore, these indices combined provide a more reliable and comprehensive view of

microbial community structure and its potential links to soil nutrient cycling. In addition, we have added some detailed description about this in Data analyses (L259-263).

3. It would be helpful to mention the perceived function of the different genes that were measured. For example, in the paragraph at L198 and in Figs. 5-7.
Response: Thank you for your suggestion. We have added some details about perceived function of the different genes in the methods (L222-225) and Figs. 5-7 sections to make it more readable (L384-386, L388-389, L400-403).

4. I think that there should be a discussion of why there was higher TP in PPs than MPs and whether trees in MPs and PPs might differ in whether they are limited by N vs. P.
Response: Thanks for pointing this out. Detailed descriptions were added in the Discussion section (L472-478).

5. The introduction and discussion would benefit from discussing mixed plantations between N-fixing and non-fixing trees in general. How representative are *Eucalyptus-Acacia* plantations of mixed plantations elsewhere?
Response: Thank you for your valuable suggestion. We have carefully re-checked the Introduction and Discussion sections and added more relevant content of mixed plantations between N-fixing and non-fixing trees (L118-130).

6. The direction of causality is unclear. Throughout the manuscript, the authors argue that microbial diversity, structure, complexity promote P transformation. However, sentences such as that on L68-70 suggest causality is in the other direction.
Response: Thank you for your suggestion. We have rephrased the sentence to avoid the confusion (L72-75).

7. The manuscript should be checked for typos and grammar. There are many instances of minor mistakes.
Response: We have carefully checked the entire manuscript and make appropriate about the organization and language of the content to make it more readable.

**Specific comments:**
1. Title: I would change to: "Soil microbial diversity and network complexity promote

phosphorus transformation: A case of long-term mixed-species plantations of *Eucalyptus* with a nitrogen-fixing tree species"

Response: Changed.

2. L24-26: Clarify that the study was in both PPs and MPs. The sentence makes it sound like the study was just done in PPs.

Response: Specified (L25-28).

3. L30: The two soil layers tested should probably be mentioned before reporting specific results for one of them.

Response: Specified (L28).

4. L63: "soil health" is a vague statement. Be more specific.

Response: Specified (L62-65).

5. L95: This sentence states that N content influences soil pH. Typically, the direction is one where an increase in N content lowers soil pH. The results show that pH however increased, which I found surprising. Although the discussion has a few lines on why, it may be good to address the hypothesized direction of change somewhere in the introduction.

Response: Thank you for your valuable suggestion. We have added some detailed description (L72-75).

6. L99: Change the part of the sentence that follows the comma to "thereby accelerating nutrient cycling and improving soil fertility"

Response: Corrected (L105-106).

7. L106: It is unclear what is meant by "soil nutrient effectiveness".

Response: Thanks for pointing this out. We have rephrased the sentence to avoid the confusion (L112-116).

8. L111: Replace "fewer" with "less or no"

Response: Changed (L235).

9. L117: This might be a good time to mention the N-fixing tree species that is used in the MPs.

Response: We have revised it as suggested (L125-130).

10. L125-126: I am unsure of what is meant by "along with genes associated with N and P cycling".

Response: Specified (L145-146).

11. L262-263: Clarify that the increase was in going from PPs to MPs.
Response: Corrected (L290-291).

12. L305: Can you explain by what metric pH is the most important regulator? It is not immediately clear from looking at Figure 3b.
Response: Thank you for your comment. The soil physicochemical properties influencing the variations of dominant microorganism phyla were identified by using redundancy analysis (RDA). The sequential selection process of RDA was used to identify the drastically distinguishing variables for soil physicochemical properties and specific microorganism phyla. Significant variables ($P < 0.05$) were employed in subsequent analysis. In our study, the value of pH ($F = 4.3$, $P = 0.003$) had the greatest impact compared to other factors ($P > 0.05$) (L330-335, L339).

13. L376: Please provide a number for the "high goodness of fit."
Response: Added (L427).

14. L450-451: Having actinobacteria in this sentence is misleading. Actinorhizal plants form N-fixing symbioses with Frankia, which are actinobacteria. However, Acacia is not an actinorhizal N fixer. Instead, Acacia forms N-fixing symbioses with Rhizobia, which are Proteobacteria.
Response: Corrected (L515-518).

15. Table 1: Clarify whether the +/- refers to the standard deviation or the standard error.
Response: Clarified (L298).

16. Table 2: Bacteria is misspelled.
Response: Corrected.

17. Figure 1: In the caption mention the threshold p value (my guess is p < 0.05) that determines whether differences between treatments are significant or not.
Response: Added (L313-314).

18. Figure 4: The Zi-Pi plots have the connectors (high among module connectivity) and module hubs (high within module connectivity) switched in the legend. Also, it is not clear what is meant by "node color node size" in the caption.

Response: Corrected (L364-367).

19. Figure 9: The caption appears to explain 9a, but not 9b.

Response: Thanks for pointing this out. We added some detailed descriptions and made it clear and specific (L440-446).

---

## Author Response (AR2)

Dear Editors:

Thank you very much for giving our chance for further revision, and your comments along with the reviewers' ones regarding our manuscript entitled "Soil microbial diversity and network complexity promote phosphorus transformation – A case of long-term mixed plantations of *Eucalyptus* and nitrogen-fixing tree species" (ID: egusphere-2024-3456). We have carefully studied your comments and made thorough revisions which have taken full considerations of your comments and suggestions.

The detailed responses for each of your comments are as follows:

1. Title and L35: Add "a" before "nitrogen-fixing tree species".

Response: Added (L2 and L27).

2. L109-132: I think much of this section should be separated into its own paragraph, separate from the last one that lists the specifics of this study, the questions, and the hypotheses (which are now clear). Additionally, I would suggest some editing of the added text here as there is some repetition (i.e., the sentence on L 127-130 repeats some information from L118-122).

Response: Thanks for pointing these out. We have carefully checked the entire paragraph again and removed L127-130 which repeats some information from L118-122.

3. L113-115: Can "reductions in soil nutrient effectiveness (e.g., the availability of nutrients such as N, and P in forms that can be absorbed and utilized by plants)" be changed to "… reductions in plant-available soil nutrients…"

Response: Changed (L114-115).

4. L121-122: Epihov et al. 2021 is an inappropriate citation for this sentence. It reports the effects of N fixers on mineral nutrients in regrowing tropical forests, not the role of N fixers on N uptake and woody production in Eucalyptus plantations.

Response: Deleted.

5. L125-127: Can you provide citations for this sentence?The last paragraph of the introduction should probably list the N-fixing tree species used in the study (Acacia mangium). There are many N-fixing tree species used in agroforestry, and they likely vary in their effects on neighboring non-fixing trees.

Response: L125-127 have been removed due to it is redundant / unnecessary sentence here. Meanwhile, we have listed the N-fixing tree species used in the study (*Acacia mangium*) in the last paragraph of the Introduction (L129-135).

6. L174-175: Can you provide citations for the previous studies being referred to?

Response: Added (L180-181).

7. L286: When reading Henseler and Sarstedt 2013 I could not find a statement stating that the goodness of fit index (from Tenenhaus et al. 2004) with a value > 0.7 indicated a good model fit. I could not find a "benchmark" that lists a good model fit.

Response: Thanks for pointing these out. We have carefully checked the related references throughout and found there were some citation errors of *Data analyses*. Further, we have corrected it in the revised manuscript (L293).

8. L290: Change "Significant" to "Significantly".

Response: Changed (L298).

9. L339: If I understand the response to my previous comment (first round of reviews) correctly, the authors are stating that pH is the most important regulator because it has the lowest p value. The p value cannot be used to determine how important a regulator (pH) is relative to other regulators (it doesn't directly map onto how much of the variance is explained by that predictor). I believe you can however use the length of the arrows to determine which regulators are more important than others.

Response: Thanks for the comments. We have carefully checked the entire paragraph and made appropriate (L340-347).

10. L427-428: The goodness of fit (Tenenhaus et al. 2004; Hensler and Sarstedt 2013) is similar to, but a bit different than the proportion of variance explained. Instead, it is the geometric average of the "average communality", i.e., the average proportion of variance explained when regressing the reflective indicators on their latent variables (the measurement model quality), and the mean block $R^2$.

Response: Thanks for pointing these out. We have carefully checked the related data analyses throughout and made necessary modification to avoid confusion (L433-436).

11. L515: It is unclear what is meant by "significant phylum".

Response: Thanks for pointing this out. We have corrected the description to avoid confusion (L523-525).

12. L517: Rhizobia are not Actinobacteria.
Response: Corrected (L525).

Once again, we would greatly appreciate Editors/Reviewers' comments and suggestions, and we believe this revision is much improved as a result of our modifications. Please let us know if there is anything else we can do to help the review process.

Thank you for your time and consideration.

Yours sincerely,

Xueman Huang